# Implicit Turn-Wise Policy Optimization for Proactive User-LLM Interaction

**Haoyu Wang** [1]  **Yuxin Chen** [2]  **Liang Luo** [2]  **Buyun Zhang** [2]  **Ellie Dingqiao Wen** [2]  **Pan Li** [1]

## Abstract

Multi-turn human-AI collaboration is fundamental to deploying interactive services such as adaptive tutoring, conversational recommendation, and professional consultation. However, optimizing these interactions via reinforcement learning is hindered by the sparsity of verifiable intermediate rewards and the high stochasticity of user responses. To address these challenges, we introduce Implicit Turn-wise Policy Optimization (ITPO). ITPO leverages an implicit process reward model to derive fine-grained, turn-wise process rewards from sparse outcome signals. Unlike volatile token-level rewards, these turn-level signals exhibit superior robustness and may utilize a normalization mechanism to further enhance training stability. We evaluate ITPO across three representative multi-turn collaborative tasks: math tutoring, document writing, and medical recommendation. Empirical results demonstrate that ITPO, when combined with PPO, GRPO, or RLOO, consistently achieves improved convergence than existing baselines. Elaborate trajectory analysis confirms that ITPO infers turn-wise preferences that are semantically aligned with human judgment. Code is publicly available at https://github.com/Graph-COM/ITPO.

## 1. Introduction

Human-AI collaboration is revolutionizing the societal landscape (Ziems et al., 2024; Li et al., 2023), ranging from enhancing personalization in recommendation systems and education (Zhu et al., 2024; Zhao et al., 2024) to the democratization of specialized expertise in healthcare and law (Peng et al., 2023; Liu et al., 2023). Realizing this potential necessitates a paradigm shift in large language models (LLMs): moving beyond reactive instruction following toward multi-turn interactions, where agents must proactively resolve ambiguity (Wu et al., 2025), decompose complex goals (Wei et al., 2022; Jaech et al., 2024), and iteratively refine their outputs (Chen et al., 2024). However, existing LLMs suffer from performance degradation in long-horizon interactions (Laban et al., 2025), making the effective alignment for multi-turn User-LLM collaboration via reinforcement learning (RL) an urgent yet open problem.

The core challenge lies in the supervision sparsity. As outcome rewards typically materialize until conversation concludes, relying only on these delayed signals often leads to poor sample efficiency and spurious solutions (Sutton et al., 1998). Synthesizing fine-grained dense process rewards offers a potential solution, while existing approaches such as value estimation (Konda & Tsitsiklis, 1999; Schulman et al., 2017) or process reward models (PRMs) (Ng et al., 1999) fall short under the multi-turn collaborative setting. On one hand, training value models to estimate process rewards is empirically difficult (Yuan et al., 2024; Guo et al., 2025; Ahmadian et al., 2024), particularly given the high variance of user dynamics that hinders convergence. On the other hand, current PRM pipelines lack scalability for multi-turn online RL (Gao et al., 2023b), relying either on labor-intensive human annotation (*e.g.,* $800,000$ labels for math problems (Lightman et al., 2023)), or Monte Carlo roll-outs with 8-10 times more sample complexity (Wang et al., 2024a; Kazemnejad et al., 2024). Moreover, unlike structured reasoning or tool-execution tasks with verifiable step-level labels (Zeng et al.; Zhao et al., 2025; Uesato et al., 2022), defining the quality of an intermediate conversational turn is often ambiguous, requiring costly task-specific rubric design (Yu et al., 2025). As to off-the-shelf LLM judges, they introduce prohibitive latency for online RL optimization (Yu et al., 2025; Lee et al., 2023) and often suffer from evaluation bias (Zheng et al., 2023).

Recently, implicit process reward models (implicit PRMs) (Cui et al., 2025; Yuan et al., 2024; Rafailov et al., 2024) have emerged as a scalable alternative, deriving dense token-level process rewards in single-turn tasks directly from outcome rewards without manual annotation or auxiliary sampling. However, this token-level granularity often induces high variance and overfitting (Razin et al., 2025), while lacking semantic interpretability due to its excessive resolu-

---

[1]Georgia Tech. <haoyu.wang@gatech.edu> [2]Meta AI. Work done during internship.
Correspondence to: Pan Li <panli@gatech.edu>.

*Proceedings of the 43rd International Conference on Machine Learning*, Seoul, South Korea. PMLR 306, 2026. Copyright 2026 by the author(s).

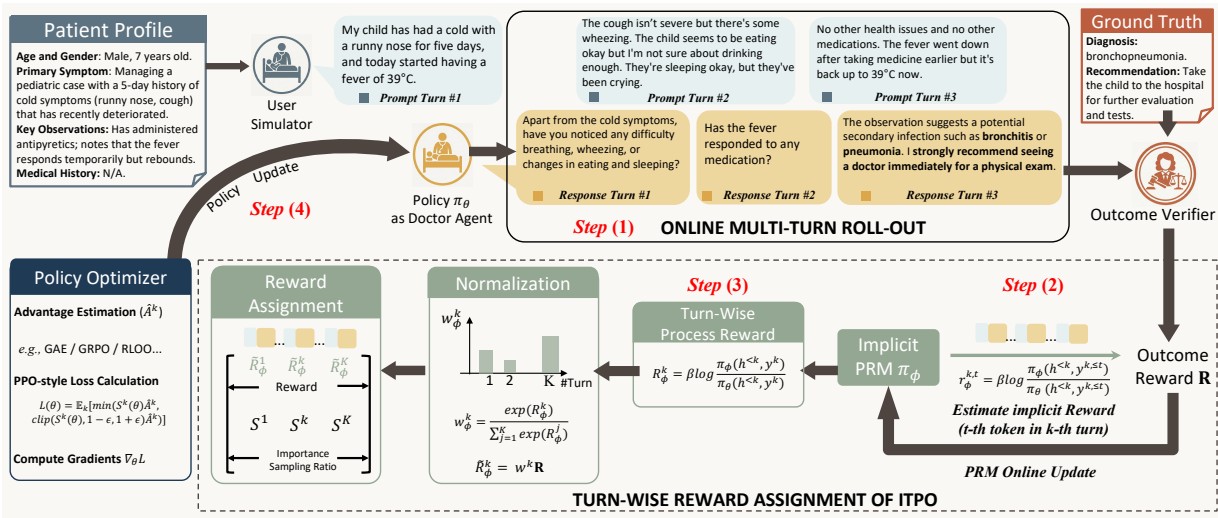

*Figure 1.* The overall framework of ITPO for User-LLM interactions with a real case example in Medical Recommendation. The optimization loop consists: of (1) Multi-turn online roll-out, (2) implicit PRM update with outcome rewards and token-level process reward estimation, (3) Turn-wise process reward aggregation and normalization, (4) Advantage estimation and policy optimization.

tion. In contrast, within multi-turn User-LLM interactions, each turn serves as a natural atomic unit for semantic planning. This level of granularity is inherently interpretable, which may facilitate monitoring by domain experts and practitioners and algorithmically reduce variance by aggregating signals across all tokens within a turn. Motivated by these insights, we propose Implicit Turn-wise Policy Optimization (ITPO), with our closed-loop online optimization framework illustrated in Figure 1. Crucially, we introduce a normalization mechanism to calibrate the learned process rewards, thereby further enhancing training stability. ITPO integrates seamlessly with various advantage functions, such as PPO (Schulman et al., 2015), GRPO (Shao et al., 2024), and RLOO (Ahmadian et al., 2024; Kool et al., 2019), to facilitate PPO-style (Schulman et al., 2017) policy updates.

We evaluate ITPO on three representative multi-turn collaborative tasks, namely Math Tutoring, Document Writing and Medical Recommendation. Analysis on training dynamics confirms that ITPO learns stable and interpretable turn-wise preference, closely aligning with human judges. When integrated with PPO (Schulman et al., 2017), RLOO (Ahmadian et al., 2024; Kool et al., 2019) or GRPO (Shao et al., 2024) advantage estimators, the turn-wise rewards provided by ITPO consistently outperform sparse outcome-based baselines as well as reward shaping methods such as PRIME (Cui et al., 2025) and LLMs-as-a-Judge (Yu et al., 2025).

## 2. Related Works

### 2.1. LLM Alignment Towards Multi-Turn Dynamics

While existing post-training based on supervised fine-tuning (Longpre et al., 2023) and RL (Schulman al.,

2017; Rafailov et al., 2023; Shao et al., 2024) prove effective for reactive instruction following (Ouyang et al., 2022), the models struggle in multi-turn interactions (Laban et al., 2025). We categorize previous efforts to mitigate this gap according to two distinct application scenarios: the reasoning and the interaction scenarios.

**Reasoning Dynamics: Goal Decomposition and Tool Use.**
The multi-step reasoning dynamics involve complex task decomposition, ranging from advanced problem-solving (Guo et al., 2025; Ouyang et al., 2023; Zou et al., 2026b;a), long-horizon gaming (Wang et al., 2025a;b), to agentic tool use (Jin et al., 2025; Gao et al., 2023a; Huang et al., 2025; Pan et al., 2023). Trajectory-level standard PPO optimization (Schulman et al., 2017) serves as fundamental baselines (Wang et al., 2025a; Guo et al., 2025; Zhou et al., 2025). To address the reward sparsity (Sutton et al., 1998), Lightman et al. (2023); Uesato et al. (2022) train process reward models (PRMs) with up to 800k human annotated labels to evaluate intermediate math reasoning steps, yielding significant performance gains, while Wang et al. (2024a) construct step-level labels via Monte-Carlo sampling. In specific tasks, Feng et al. (2025a) perform baseline estimation by clustering identical or similar states in finite environments, Zeng et al. derive turn-level rewards from verifiable tool-use labels. Crucially, these successes rely on the verifiability of intermediate steps.

**Interaction Dynamics: Dialogue and Collaboration.**
The interaction dynamics focus on collaborative multi-turn tasks involving human users or agent partners, spanning social science simulation (Yu et al., 2025; Wang et al., 2024b), recommendation systems (Wan et al., 2025; Bao et al., 2023), and collaborative assistance (Wu et al., 2025;

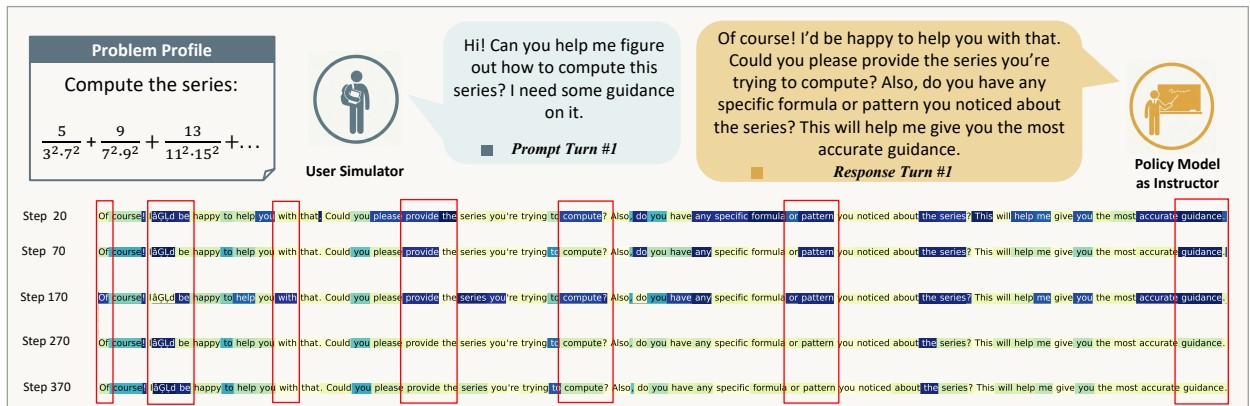

*Figure 2.* Dynamics of the relative implicit rewards on a real, fixed Math Tutoring initial prompt and its response #1 from one of the multi-turn rollout trajectories. The lower part shows the implicit process rewards assigned to each token in response #1 over reward model training, from step 20 to step 370. Darker colors refer to higher implicit scores, while lighter colors indicate lower scores. Human judges consider the response #1 to be the best among all turns, because it concisely requests the missing information. Turn-wise rewards assigned by ITPO also consistently rank response #1 the highest from step 20 onward, reflecting the importance of resolving ambiguity early in the conversation. By contrast, the token-level scores fluctuate significantly throughout training steps, as highlighted by the red boxes. Moreover, the tokens receiving high scores do not exhibit a clear semantic pattern (e.g, 'the', 'you', 'to'...).

Sun et al., 2025). Wu et al. (2025) employ vanilla PPO with forward sampling for reward estimation in multi-turn collaborative tasks. Approaches that define fine-grained turn-wise rewards have shown effectiveness. For instance, Wan et al. (2025) model categorical user belief states within a POMDP framework for simplified recommendation, while Sotopia-RL (Yu et al., 2025) leverage task-specific rigid rubrics and LLM-as-a-Judge to assign semantic turn-level rewards. However, these methods depend on restrictive task-specific priors. There is a critical need for scalable and automated algorithms to produce turn-wise rewards.

### 2.2. Implicit Process Reward Models

Implicit PRMs (Yuan et al., 2024; Cui et al., 2025; Rafailov et al., 2024) derive fine-grained token-level process rewards solely from verifiable outcome supervision. Specifically, these methods formulate the token-level process reward as the log-likelihood ratio between the reward model and the reference model, allowing the gradient from sparse outcome supervision to implicitly propagate credit to individual tokens. By eliminating the need for human process annotations, implicit PRMs offer a scalable alternative to explicit PRMs. Recently, theoretical analysis (Razin et al., 2025) suggests that while the token-level implicit process reward can effectively fit training distributions as explicit PRMs, it often suffers from poor generalization. The high variance inherent in assigning dense token-level credit from sparse outcomes creates a noisy supervision signal, highlighting the need for stable improvements. (Liu et al., 2025) demonstrate that assigning token-level implicit rewards to intermediate steps effectively improves RL training for LLMs on agentic planning and related benchmarks. Under the multi-turn interaction settings, the variance is further exac-

erbated by the stochastic user dynamics; meanwhile, the fine-grained supervision on isolated tokens lacks semantic interpretability.

**User Simulation for Evaluation and Training** User simulators have become pivotal for studying systems involving human interaction (Guo et al., 2024; Sun et al., 2024), widely applied in social science (Tang et al., 2025), economic modeling (Horton et al., 2023), and debate (Du et al., 2023). Recent evaluations (Dou et al., 2025; Naous et al., 2025) demonstrate that LLMs conditioned on detailed user profiles can closely align with human judgments and serve as reliable stand-ins for real users, enabling the training and evaluation of policies in a controlled environment.

## 3. Methodology

We present the methodology by first formalizing the problem of the multi-turn collaborative task and second the preliminary fundamentals of implicit PRM. Subsequently, we introduce ITPO, describing how it derives robust turn-wise rewards to enable scalable online policy optimization.

### 3.1. Problem Formulation

We formulate the multi-turn collaborative task as a Partially Observable Markov Decision Process (POMDP) (Sondik, 1971; Åström, 1965), defined by the tuple $\langle \mathcal{S}, \mathcal{Y}, \mathcal{T}, \mathbf{R}, \gamma \rangle$. The state space $\mathcal{S} = \mathcal{H} \times \mathcal{G}$ comprises the observable conversation history $h^k \in \mathcal{H}$ and the user's latent goal $g \in \mathcal{G}$, which is unobservable to the agent. The action space corresponds to the set of model responses $\mathcal{Y}$. At turn $k$, the agent policy $\pi_\theta$ generates a response $y^k \in \mathcal{Y}$ conditioned on $h^k$. The transition dynamics $\mathcal{T}$ are governed by the

distribution of user's responses $P_U(x^{k+1}|h^k, y^k, g)$, where the user query $x^{k+1}$ serves as the observation for the next turn. The interaction terminates after $K$ turns, yielding a trajectory $\tau = [(x^1, y^1), \ldots, (x^K, y^K)]$ with a sparse outcome reward $\mathbf{R}(\tau) \in [0, 1]$[1], representing the quality of the collaboration. Consequently, the core objective is to maximize the expected outcome reward $\mathbb{E}[\mathbf{R}]$. Given the finite horizon of dialogue tasks, discount factor $\gamma$ is set to 1. Throughout this paper, we prompt an LLM-based user simulator to approximate $P_U$ (Dou et al., 2025).

Optimizing the objective above presents two challenges. First, the distribution $P_U$ of the observation $x^k$ given the latent goal $g$ could have high variance: users with identical goals may formulate diverse queries or distinct interaction styles. The agent needs to implicitly reason about $g$ under high uncertainty. Second, accurate value estimation is challenging. The sparse reward $\mathbf{R}(\tau)$, compounded by the high variance in user dynamics, renders the estimation of $V_\psi(h^k) = \mathbb{E}_\tau[\mathbf{R}(\tau)|h^k]$ sample-inefficient. A potential solution is reward shaping (Ng et al., 1999), which decomposes the outcome $\mathbf{R}(\tau)$ into a sequence of dense, proxy rewards to provide fine-grained feedback.

### 3.2. Preliminary: Implicit Process Reward Model

Implicit PRMs (Yuan et al., 2024; Cui et al., 2025) parameterize dense, token-level process rewards via generative models, enabling online update using solely outcome rewards. Let $y^{k,t}$ denote the $t$-th token of response $y^k$ at turn $k$, the corresponding token-level process reward is defined as the scaled log-likelihood ratio between an implicit PRM $\pi_\phi$ and a fixed reference model $\pi_{\text{ref}}$:

$$r_\phi(y^{k,t} \mid h^k, x^k, y^{k,<t}) = \beta \log \frac{\pi_\phi(y^{k,t}|h^k,x^k,y^{k,<t})}{\pi_{\text{ref}}(y^{k,t}|h^k,x^k,y^{k,<t})}, \quad (1)$$

where $\beta$ is a scaling coefficient, $h^k$ is the history before turn-$k$, $x^k$ is the user query, and $y^{k,<t}$ denotes the prefix. The trajectory-level implicit reward $\mathcal{R}_\phi(\tau)$ aggregates these token-level rewards over the sequence:

$$\mathcal{R}_\phi(\tau) = \sum_{k=1}^{K} \sum_{t=1}^{|y^k|} r_\phi(y^{k,t} \mid h^k, x^k, y^{k,<t}). \quad (2)$$

Optimization aligns $\mathcal{R}_\phi(\tau)$ with the ground-truth outcome $\mathbf{R}(\tau)$ via binary cross-entropy:

$$\mathcal{L}_{\text{PRM}}(\phi) = -\mathbb{E}_{(\tau,\mathbf{R})\sim\mathcal{D}}\left[\mathbf{R}\log\sigma\left(\mathcal{R}_\phi(\tau)\right) + (1-\mathbf{R})\log\left(1-\sigma\left(\mathcal{R}_\phi(\tau)\right)\right)\right], \quad (3)$$

---

[1]Without loss of generality, while we adopt the probabilistic interval $[0, 1]$ to ground the derivation, the proposed attribution mechanism remains valid for any bounded reward scale $\mathbf{R} \in [a, b]$ via linear scaling. Furthermore, ITPO can accommodate multiple reward signals through linear aggregation (i.e., $\mathbf{R} = \sum_m \mathbf{R}^{(m)}$).

where $\sigma$ denotes sigmoid function. Invoking **Proposition 3.1** from Cui et al. (2025), the partial accumulation functions as an implicit Q-value estimator, $Q_\phi(h^k, x^k, y^{k,\leq t}) \approx \sum_{j=1}^{k-1} \sum_{p=1}^{|y^j|} r_\phi(y^{j,p} \mid \cdot) + \sum_{i=1}^{t} r_\phi(y^{k,i} \mid \cdot)$, establishing $r_\phi$ as the token-wise implicit value increment.

Standard token-level implicit PRMs face three limitations. (1) High variance for token-level reward. As visualized by the red boxes in Fig. 2, while the aggregated turn-wise score correctly identifies Response Turn #1 as the pivotal turn after just 20 updates, the internal token-level rewards remain chaotic along training steps. Also as shown in Fig. 3, turn-wise preferences stabilize rapidly, whereas token-level rewards exhibit significantly slower convergence. (2) Semantic misalignment. As in the example, high rewards fluctuate arbitrarily across functional tokens (e.g., "with", "please") during training. (3) Overfitting Issue. Theoretically the overly fine-grained supervision exacerbates the susceptibility to overfitting (Razin et al., 2025).

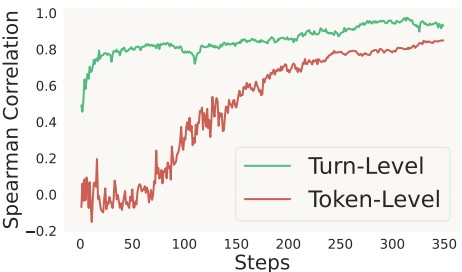

*Figure 3.* Spearman correlation of the token-level and turn-wise implicit rewards against their respective converged baselines (the averaged rewards over the final 70 optimization steps) on Math Tutoring. Turn-wise preference stabilizes quickly, while token-level rankings show slower convergence due to optimization difficulty.

### 3.3. Implicit Turn-Wise Policy Optimization (ITPO)

Given the preliminaries, we introduce ITPO and its normalized version Norm-ITPO.

**ITPO** mitigates token-level noise by aggregating the token-level evidence, where the turn-wise implicit reward $\mathcal{R}_\phi^k$ is defined as the accumulation of token-level log-likelihood ratios within the $k$-th turn:

$$\mathcal{R}_\phi^k = \sum_{t=1}^{|y^k|} r_\phi(y^{k,t} \mid h^k, x^k, y^{k,<t}), \quad (4)$$

where $|y^k|$ denotes the length of the response at turn $k$. Physically, $\mathcal{R}_\phi^k$ represents the cumulative implicit advantage of the turn. ITPO effectively mitigates the high variance and noise inherent in individual token-level estimations, as illustrated in Fig. 3.

**Norm-ITPO.** While the learned trajectory-level implicit reward $\mathcal{R}_\phi(\tau)$ exhibits a positive correlation with the outcome reward $\mathbf{R}(\tau)$, the scale of such mapping suffer from

fluctuation, even on fixed roll-out trajectories (See Fig. 7). The instability poses two challenges: it introduces variance into policy updates and creates a non-stationary target that hinders value model convergence. These observations motivate Norm-ITPO, which explicitly calibrates the reward scale to ensure optimization stability.

Norm-ITPO utilizes the aggregated turn-wise scores to redistribute the global outcome reward **R**. We employ a Softmax function to compute the contribution weight $w_k$ for each turn $k$:

$$w_\phi^k = \text{Softmax}(\mathcal{R}_\phi^k/\eta) = \frac{\exp(\mathcal{R}_\phi^k/\eta)}{\sum_{j=1}^K \exp(\mathcal{R}_\phi^j/\eta)}, \quad (5)$$

where $\eta$ is a temperature hyperparameter. The normalized reward for turn $k$ is then computed as a fraction of the global outcome:

$$\tilde{\mathcal{R}}_\phi^k = w_\phi^k \cdot \mathbf{R}, \quad (6)$$

By doing normalization, Norm-ITPO not only provides inter-turn preference but also enforces scale consistency between the implicit reward and the outcome **R**, which prevents value drift and stabilizes the training dynamics. Notably, this formulation generalizes two fundamental strategies: as $\eta \to 0$, the reward is assigned solely to the most pivotal turn, a strong baseline in social simulation (Yu et al., 2025); conversely, as $\eta \to \infty$, the distribution becomes uniform, corresponding to a mean-contribution baseline where every step shares the contribution equally.

**Bayesian Interpretation of Norm-ITPO.** The redistribution mechanism can be grounded in Bayesian inference over a latent pivotal turn $Z \in \{1, \ldots, K\}$. Adopting a uniform prior $P(Z) = 1/K$ and modeling the likelihood of the implicit evidence $\mathcal{R}_\phi^k$ as a Boltzmann distribution $P(\mathcal{E} \mid Z = k) \propto \exp(\mathcal{R}_\phi^k/\lambda)$, the posterior probability of turn $k$ being the decisive factor is derived via Bayes' theorem:

$$P(Z = k \mid \mathcal{E}) = \frac{\exp(\mathcal{R}_\phi^k/\eta)}{\sum_{j=1}^K \exp(\mathcal{R}_\phi(y_j)/\eta)}, \quad (7)$$

which equals to weight $w^k$. Consequently, the assigned reward $\tilde{\mathcal{R}}_k$ represents the expected credit attributed to turn $k$ under the posterior belief.

Overall, ITPO and Norm-ITPO provide fine-grained turn-wise process rewards allocation that exhibits self-consistency, stability, and interpretability. Refer to Section 4.3.2 for a comprehensive empirical validation.

### 3.4. Policy Optimization

The derived turn-wise rewards can be integrated with standard advantage estimators. For the $k$-th turn in the $i$-th sample, the advantage $\hat{A}^{i,k}$ is computed as follows:

**RLOO (Ahmadian et al., 2024):** This method uses a leave-one-out baseline to reduce variance without a parametric value model. The advantage is estimated as $\hat{A}_{\text{RLOO}}^{i,k} = \tilde{\mathcal{R}}_\phi^{i,k} - \frac{1}{G-1}\sum_{j\neq i}\tilde{\mathcal{R}}_\phi^{j,k}$, where $G$ denotes the number of roll-out trajectories in the group sharing same prompts.

**GRPO (Shao et al., 2024):** To stabilize training, GRPO standardizes rewards based on group statistics. The advantage is computed as $\hat{A}_{\text{GRPO}}^{i,k} = (\tilde{\mathcal{R}}_\phi^{i,k} - \mu)/(\sigma + \epsilon)$, where $\mu$ and $\sigma$ represent the mean and standard deviation of the group rewards $\{\tilde{\mathcal{R}}_\phi^{j,k}\}_{j=1}^G$, respectively.

**Learned Value Model (Schulman et al., 2015):** Following the PPO standard, this approach employs Generalized Advantage Estimation (GAE) with a separate critic network $V_\psi$. The advantage is defined as $\hat{A}_{\text{PPO}}^{i,k} = \sum_{l=0}^{K-k-1}(\gamma\lambda)^l \delta^{i,k+l}$, where the term $\delta^{i,t} = \tilde{\mathcal{R}}_\phi^{i,t} + \gamma V_\psi(h^{i,t+1}) - V_\psi(h^{i,t})$ represents the turn-level Temporal Difference (TD) error.

**Loss Function.** The policy $\pi_\theta$ is updated via PPO loss (Schulman et al., 2017), with turn-level importance sampling ratio defined as $\rho^{i,k}(\theta) = \frac{\pi_\theta(y^{i,k}|h^{i,k})}{\pi_{\text{old}}(y^{i,k}|h^{i,k})}$. The objective is to maximize the clipped surrogate objective:

$$\mathcal{L}_{\text{PPO}}(\theta) = \mathbb{E}\left[\sum_{k=1}^K \min\left(\rho^{i,k}\hat{A}^{i,k}, \text{clip}(\rho_{i,k}, 1-\epsilon, 1+\epsilon)\hat{A}^{i,k}\right)\right]. \quad (8)$$

This formulation shifts optimization from tokens to turns. By enforcing the sampling ratio on the entire turn, ITPO preserves semantic coherence and avoids the disruption of joint probability dependencies caused by token-level clipping (Zheng et al., 2025).

## 4. Experiments

### 4.1. Settings and Implementation

#### 4.1.1. DATASETS AND METRICS

Experiments are conducted on three representative multi-turn collaborative use cases: education (math tutoring), content generation (document writing) and knowledge-intensive recommendation (medical recommendation) (Tamkin et al., 2024; Sun et al., 2025; Achiam et al., 2023; Touvron et al., 2023). For details, see Appendix. A.1.

**Document Writing** is representative for iterative content generation. The core challenge is to ensure alignment with the user's latent intent through iterative refinement. Each instance is anchored by a ground-truth *goal document* and a high-level summary. The user simulator is prompted with the summary, utilizing it to guide the LLM's drafting. Performance is quantified with **BLEU** score, measuring the semantic overlap between the generated draft and the goal. We adopt the protocol from Wu et al. (2025), utilizing a dataset of 500 curated articles sourced from Medium.

**Math Tutoring** evaluates the capability in handling under-specified queries. In real-world tutoring, students often

provide incomplete problem descriptions, requiring the tutor to actively solicit information through iterative interactions. In our setup, each instance consists of a math problem and its ground truth solution. The user simulator holds the full problem but reveals them partially and deliberately to the LLM. Task success is evaluated by the **Accuracy (ACC)** of the answer, as verified by an LLM judge. The dataset comprises 500 problems from MATH (Hendrycks et al., 2021), following the configuration in Wu et al. (2025).

**Medical Recommendation** targets knowledge-intensive recommendation. We model this as a diagnostic consultation where a patient presents symptoms to a doctor. The LLM performs symptom elicitation to reach a diagnosis and a recommendation rooted in expert knowledge. The user simulator is conditioned on a patient profile. Performance is assessed via a hierarchical **Medical Reward** assigned by an expert LLM judge. Score 0 is assigned for incorrect diagnoses; scores 1–3 for correct diagnoses paired with suboptimal or hallucinated recommendations; and scores 8–10 for accurate diagnoses. The dataset consists of 550 samples from the MTMedDialog benchmark (Feng et al., 2025b).

### 4.1.2. BASELINES FOR REWARD SHAPING

We compare ITPO against five baselines with granularity from trajectory to token-level. Details in Appendix. A.2.1.

**Share along Trajectory:** The standard outcome supervision baseline that underpins RLOO (Ahmadian et al., 2024) and GRPO (Shao et al., 2024). It broadcasts the final outcome reward $\mathbf{R}$ across all tokens in the trajectory. The method ignores the temporal structure of the interaction, treating every token as an equal contribution.

**Uniform Decomposition:** distributes the outcome reward uniformly with randomness. The weights are modeled with Dirichlet distribution $\mathbf{w} \sim \text{Dirichlet}(\mathbf{1})$, with turn-wise rewards defined as $R^k = w_k \cdot \mathbf{R}$. This provides a dense reward baseline where every interaction turn contributes a uniform and positive marginal gain towards the final outcome.

**Value Model:** learns a value function $V_\psi$ to provide dense, token-level feedback (Schulman et al., 2017). The reward signal is derived from the temporal difference (TD) error: $\delta^{k,t} = \gamma V_\phi(h^{<k}, y^{k,\leq t}) - V_\phi(h^{<k}, y^{k,<t})$ (Schulman et al., 2015). $\gamma$ is set as 1 due to the finite nature of the task. While this offers fine-grained granularity, training a critic network may suffer from high variance and slow convergence.

**LLM-as-Judge:** leverages the semantic reasoning capabilities of an external judge, which is shown effective in social tasks (Yu et al., 2025). The judge is prompted to analyze the conversation and assign a normalized scalar weight $w^k$ to each turn resulting in a shaped reward $R^k = w^k \mathbf{R}$. Given the prohibitive cost and latency of API-based models in on-line RL, we employ Qwen2.5-14B-Instruct, which demonstrates comparable capability to GPT-4o-mini (Achiam et al., 2023) in the report (Yang et al., 2025b).

**Implicit PRM:** (Yuan et al., 2024) formulated as $r_\phi^{k,t} = \beta \log \frac{\pi_\phi(\cdot)}{\pi_{\text{ref}}(\cdot)}$. While this method serves as the atomic evidence for ITPO, we include it as a baseline to explicitly demonstrate the limitations of token-level supervision in multi-turn interactions, which often suffer from high variance and susceptible to overfitting (Razin et al., 2025).

### 4.1.3. IMPLEMENTATION

**Backbone** We employ Qwen2.5-14B-Instruct as the user simulator. Main evaluations use Qwen2.5-3B-Instruct (Yang et al., 2025b) as policy. To verify the generalizability of our method, we provide additional results with Qwen3-4B (Yang et al., 2025a) and Qwen2.5-7B-Instruct (Yang et al., 2025b) as policy. The implicit PRM is initialized with the same parameters as the policy. We perform full-parameter fine-tuning on the policy across all experiments.

**Reward Design** To discourage verbosity without compromising task performance, the optimization objective is formulated as the combination of the task-specific metric and a length regularization term. The objective is to maximize a score $S = \mathbf{R} - \gamma \times N$, where $\mathbf{R} \in [0, 1]$ represents task-specific metric as defined in Section. 4.1.1 and $N$ is the token count of the model output, we set $\gamma$ as 5e-6 across all the tasks.

**Reward Model Training** If training PRM is required, both the baselines and ITPO are trained with the outcome metric $\mathbf{R}$ as objective. The token amount regularization is explicitly calculated and assigned to individual turns.

**Policy Optimization** In all experiments, we employ the PPO loss defined in Equation. (8), with a KL penalty coefficient of 1e-3 by default. We evaluate three advantage estimators: RLOO (Kool et al., 2019), GRPO (Shao et al., 2024), and GAE (Schulman et al., 2015), calculated using equations in Section. 3.4. Details in Appendix. A.2.2.

### 4.2. Main Results

We integrate reward allocation methods as introduced above and pair Qwen2.5-3B-Instruct (Yang et al., 2025b) policy with RLOO (Kool et al., 2019), GRPO (Shao et al., 2024) and PPO that learns Value Model (Schulman et al., 2017; 2015) advantage functions across three tasks. The evaluation results are in Table. 1, with training dynamics in Fig. 4. The dynamics maintain consistent observation as the evaluation.

**ITPO and Norm-ITPO outperform the baselines.** According to Table. 1, both ITPO and Norm-ITPO consistently outperforms the other reward allocation baselines across the tasks and advantage estimators. For in-

| Advantage Estimator | Credit Assignment | Math Tutoring | | | Doc Writing | | | Medical Recommendation | | |
|---|---|---|---|---|---|---|---|---|---|---|
| | | Accracy↑ | Token(k)↓ | Score↑ | Bleu↑ | Token(k)↓ | Score↑ | Diagnosis↑ | Token(k)↓ | Score↑ |
| | Vanilla | 25.81 | 1.70 | 17.29 | 24.97 | 2.40 | 12.93 | 25.52 | 0.28 | 24.08 |
| PPO | VM Only | 26.81 | 1.36 | 20.06 | 40.57 | **0.84** | 36.37 | 58.41 | 0.72 | 54.77 |
| | VM+PRIME | 22.63 | 1.54 | 14.92 | 41.87 | 2.42 | 29.77 | 52.07 | **0.48** | 49.73 |
| | VM+ITPO (Ours) | 26.38 | 0.94 | 21.51 | 42.52 | 1.02 | 37.44 | 58.35 | 0.68 | 54.87 |
| | VM+Norm-ITPO (Ours) | **27.15** | **0.74** | **23.44** | **45.16** | 0.96 | **40.12** | **62.44** | 0.60 | **59.44** |
| RLOO | Trajectory-Level | 29.06 | 1.44 | 21.82 | 37.35 | **0.42** | 35.34 | 65.12 | 0.78 | 61.22 |
| | LLM as Judge | 28.87 | 1.24 | 22.59 | 42.34 | 1.04 | 37.20 | 66.77 | 0.94 | 62.03 |
| | Uniform Decompose | 30.00 | 1.18 | 24.01 | 37.81 | 0.86 | 33.45 | 63.15 | **0.46** | 60.82 |
| | PRIME | 29.75 | 1.54 | 21.54 | 40.95 | 0.74 | 37.21 | 61.42 | 1.12 | 55.90 |
| | ITPO (Ours) | 29.06 | **0.58** | 26.08 | 44.59 | 1.08 | 39.12 | 68.43 | 0.98 | 63.49 |
| | Norm-ITPO (Ours) | **32.50** | 0.62 | **29.33** | **44.83** | 1.04 | **39.59** | **69.24** | 0.62 | **66.14** |
| GRPO | Trajectory-Level | 26.37 | 0.98 | 21.41 | 39.33 | **0.36** | 37.28 | 66.41 | 0.94 | 61.67 |
| | LLM as Judge | 30.62 | 1.60 | 22.57 | 42.47 | 1.06 | 37.15 | 63.12 | 1.12 | 60.56 |
| | Uniform Decompose | 26.62 | 1.30 | 20.71 | 38.47 | 0.48 | 36.00 | 61.95 | 0.66 | 58.64 |
| | PRIME | 30.19 | 1.13 | 24.53 | 39.54 | 1.10 | 33.99 | 56.36 | 0.65 | 53.12 |
| | ITPO (Ours) | 26.37 | **0.58** | 23.51 | **44.60** | 0.94 | **39.92** | 67.61 | 0.91 | 63.07 |
| | Norm-ITPO (Ours) | **31.12** | 0.84 | **26.83** | 42.78 | 0.98 | 37.86 | **71.74** | **0.62** | **68.66** |

*Table 1.* Evaluation results of fine-grained reward attribution strategies, with task-specific metrics (Accuracy, BLEU, Diagnosis Score), generation cost (Tokens), and overall utility scores. **Bold** values denote the best performance within each group, Underlined denotes the second. 'Vanilla' denotes the unaligned backbone policy, 'VM' (Value Model) denotes the inclusion of a parameterized value model.

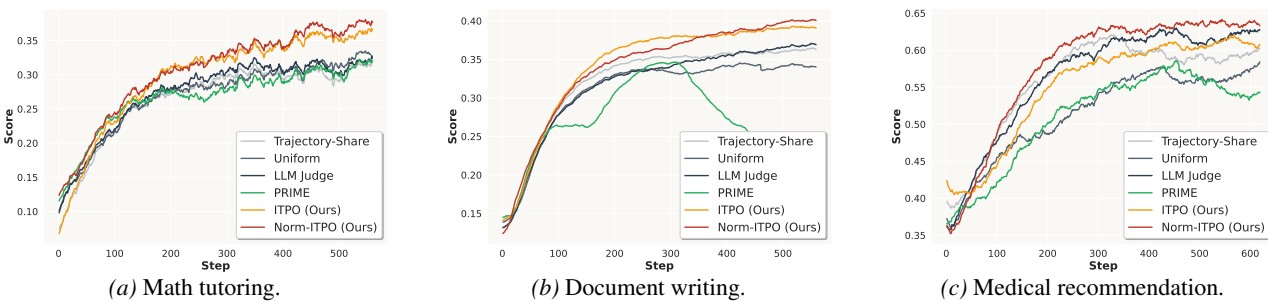

| (a) Math tutoring. | (b) Document writing. | (c) Medical recommendation. |

*Figure 4.* The training curve of reward attribution methods with the RLOO advantage estimator.

stance, Norm-ITPO improves the trajectory-level sparse baseline adopted by vanilla RLOO by $34.4\%, 12.0\%, 8.0\%$ on Math Tutoring, Doc Writing and Medical Rec tasks, while improves vanilla GRPO by $25.3\%, 1.6\%, 11.3\%$ respectively. Training an extra value model based on the turn-wise reward assigned by Norm-ITPO also brings an extra $16.7\%, 10.3\%, 8.5\%$ performance gain than that trained on only the outcome reward.

While Norm-ITPO outperforms ITPO across the settings, the performance gap is notably more pronounced when employing a value model (i.e., PPO) compared to the others. For example, on the Medical Recommendation task, Norm-ITPO surpasses ITPO by 8.3% in the PPO setting, whereas the margin narrows to 4.2% with RLOO. This empirical observation validates our discussion regarding scale instability in Section 3.3. Although the implicit rewards in ITPO correlate positively with outcomes, the fluctuating scale relationship (Fig. 7) creates a *non-stationary target* for the value function estimation. By explicitly enforcing consistency between the aggregated implicit reward and the outcome via Eq. 6, Norm-ITPO provides a stable regression target, leading to the substantial performance gains observed

in the PPO experiments.

**Comparison among the baselines.** Among the other baselines, LLM as Judge can stably improve the performance of vanilla baselines with sparse outcome rewards, showing that the semantic information leveraged by off-the-shelf LLMs can provide fine-grained help. But there is still a gap towards the methods that update the reward model online with the training data. PRIME (Cui et al., 2025) does not show stable convergence on the Doc Writing task, and generally achieves comparable performance with vanilla baselines, mainly due to the high variance in token-level supervision. Uniform Decompose can only achieve comparable performance with the vanilla baselines.

**Comparison among advantage estimation functions.** PPO that trains an extra value model does not help the performance on these tasks. The main reason is the same as those revealed in prior works (Shao et al., 2024), the convergence towards the real expectation distribution is challenging.

## 4.3. Effectiveness Analysis for Implicit PRM

The effectiveness unfolds into two stages. In the first hierarchy, we validate the model's ability to distinguish trajectory-level quality within response groups. Secondly, we examine the internal turn-wise reward allocation, specifically verifying the consistency, stability and semantic interpretability.

### 4.3.1. TRAJECTORY-LEVEL REWARD PREDICTION

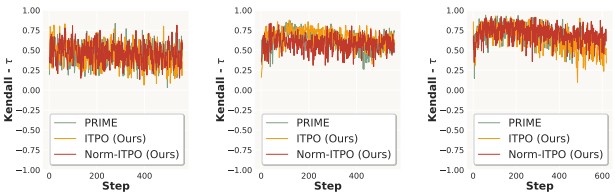

*(a)* Math Tutoring    *(b)* Doc Writing.    *(c)* Medical Rec.

*Figure 5.* Kendall-$\tau$ between PRM and outcome reward w/ RLOO.

We begin by verifying whether the implicit $\mathcal{R}_\phi$ serves as an accurate proxy for the ground-truth outcome $\mathbf{R}$ at the trajectory level. For each input prompt $x$ in a sampled mini-batch $\mathcal{B}$, we generate a group of roll-outs and compute the Kendall-$\tau$ correlation between the cumulative implicit reward $\mathcal{R}_\phi$ and $\mathbf{R}$ within each group, and report the correlation averaged over the mini-batch. As in Fig. 5, the implicit PRM of all the three (PRIME (Cui et al., 2025), ITPO, Norm-ITPO) demonstrates rapid convergence, achieving a stable Kendall-$\tau$ correlation of 0.5–0.75 with the outcome reward. This indicates that the implicit PRMs effectively learns to discriminate trajectory quality within the roll-out groups with high efficiency, providing a reliable signal along trajectories.

### 4.3.2. TURN-WISE REWARD ALLOCATION

We further analyze the stability, consistency, and semantic interpretability of the learned turn-wise rewards by ITPO and Norm-ITPO. We employ a fixed probe set consisting of 400 multi-turn trajectories (8 roll-outs per prompt across 50 validation prompts). Throughout the training process, we perform periodic inference on this static set, tracking the dynamics of the assigned implicit rewards.

**Turn-Wise Preference: Stability & Consistency**

As visualized in Fig. 6, the implicit turn-wise reward allocation exhibits rapid convergence, stabilizing quickly during the early stages of training. Once settled, the rewards display high temporal stability with respect to the time-averaged converged state, indicating that the model retains its learned preferences without undergoing abrupt shifts (Left panels). In addition, the strong mutual correlation between ITPO and Norm-ITPO demonstrates that the learned preference structure is consistent and intrinsic; the model discovers the same underlying reward signal despite differences in random initialization and stochastic roll-out trajectories. Finally, the consistently high correlation between adjacent

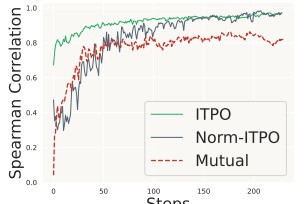 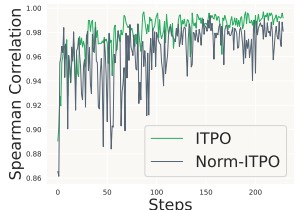

*(a)* Stability & Convergence    *(b)* Self-Consistency

*Figure 6.* **Left** tracks the Spearman correlation of turn-wise implicit rewards between each training step and the converged reward allocation (calculated as the average over the last 70 steps). Additionally, the mutual Spearman correlation between ITPO and Norm-ITPO demonstrates that both methods learn a highly consistent preference structure independent of initialization. **Right** measures the correlation between adjacent training steps.

training steps (Right panels) demonstrate that the turn-wise implicit reward optimization is smooth and free from significant oscillation.

**Scale Stability between Implicit and Outcome Rewards**
For each roll-out group that shares same initial prompts, we perform linear regression on the trajectories between their implicit trajectory-level reward $\mathcal{R}_\phi(\tau)$ and the outcome reward $\mathbf{R}$ to quantify the positive correlation. The slopes of the regression are displayed in Fig. 7, where each curve represents the slope change of the group along training steps. Although the implicit reward exhibit strong positive correlation with the outcome reward, their mapping scale fluctuates, which may introduce variances into policy updates and hinder the value model convergence.

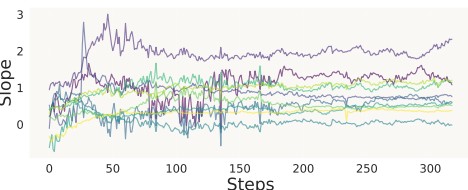

*Figure 7.* Slopes of the mapping between trajectory-level implicit $\mathcal{R}_\phi(\tau)$ and outcome $\mathbf{R}$ on fixed sets of roll-outs trajectories from the validation set in Doc Writing task. Each curve represents the slope of a roll-out group with same initial prompts during training.

**Semantic Interpretability** We further examine the interpretability of implicit turn-wise rewards by randomly sampling 32 instances from the fixed probe set. Three human experts were tasked with identifying the 'best' and 'worst' turns within each trajectory based on response quality. To ensure a high-quality ground truth, we adopted a consensus-based annotation protocol, where independent judgments were aggregated and discrepancies were resolved through expert adjudication. We evaluate the implicit turn-wise rewards against these human labels across 64 decision points (32 best and 32 worst turns). As a reference for state-of-the-art capability, Gemini-3.0-Pro (Team et al., 2023) (leveraging extended reasoning) achieved a near-perfect alignment

of $58/64$ matches. The learned reward models demonstrated high consistency with human preferences: ITPO matched $47/64$ ($73.4\%$) and Norm-ITPO matched $48/64$ ($75\%$) of the labels. These results significantly outperform the random permutation baseline, which yields an expected agreement of only $16/64$ when accounting for the constraint that best and worst turns must be distinct. This confirms that the implicit PRM captures meaningful semantic preferences rather than relying on stochastic artifacts. All examples are available in Appendix. A.5.

### 4.4. Applicability to other Base Policy Models

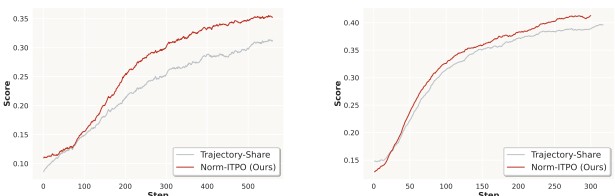

*Figure 8.* Qwen3-4B (left) and Qwen2.5-7B (right) on Doc Writing.

To verify the generalizability of ITPO across different model architectures, we conduct additional experiments using additional policy backbones. We employ Qwen3-4B-Instruct-2507 (Yang et al., 2025a) and the larger Qwen2.5-7B-Instruct (Yang et al., 2025b) as base models on the Document Writing task with RLOO advantage estimator. As illustrated in Fig. 8, Norm-ITPO consistently outperforms the baseline with shared trajectory-level outcome reward across both model sizes. The results confirm that the effectiveness is robust to variations in model scale or architecture.

### 4.5. Additional ablation studies

We further conduct additional ablation studies, including replacing the BLEU-based reward signal with an LLM-as-a-judge reward in the Doc Writing task, switching to alternative user simulator and judge models, and using performance alone without penalizing token usage. Details are in Appendix. A.4.

## 5. Conclusion and Limitations

In this paper, we introduce Implicit Turn-wise Policy Optimization (ITPO), a framework designed to address the reward sparsity challenge in multi-turn User-LLM interactions. ITPO derives fine-grained turn-wise implicit rewards with an implicit process reward model from sparse outcome rewards and may utilize a normalization mechanism to further enhance training stability. The turn-wise signals exhibit superior robustness and infer preferences within the trajectories with semantic interpretability that aligns with human judgment. ITPO is evaluated with PPO, GRPO and RLOO across Math Tutoring, Document Writing and Medical Rec-

ommendation tasks, which demonstrate that ITPO brings improvement to standard outcome-based baselines as well as existing reward allocation baselines.

**Limitations** The experiments rely on LLM-based user simulators, which although provide an efficient controlled environment, yet may not fully capture the diversity and variability of real human users. Therefore, further research on developing more capable user simulators would be valuable. Second, our method depends on LLM judges and learned reward signals, the judge might be biased and the reward improvements may not always translate perfectly to better user-perceived quality. Finally, some tasks considered in this work, such as medical recommendation, are intended primarily as challenging simulated benchmarks for knowledge-intensive interaction. They should not be interpreted as evidence of readiness for real-world high-stakes deployment.

## Acknowledgements

H. Wang and P. Li are funded partially by NSF. 2117997, NSF. 2428777, and NSF. 2239565.

We thank Deyu Zou and Yongqiang Chen for their insightful discussions and valuable suggestions through the project.

## Impact Statement

This paper presents work whose goal is to advance the field of Machine Learning. There are many potential societal consequences of our work, none which we feel must be specifically highlighted here.

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

# A. Appendix

## A.1. Task Details: Protocol and Prompts

**Math Tutoring** In the Math Tutoring dataset, the policy model has a maximum generation token size 4096 in total, with prompt size set as 1024, the conversation terminates within at most 5 turns. The system prompt for policy as assistant is as below:

**System Prompt for Policy in Math Tutoring**

```
The assistant is designed to be helpful, proactive, and highly interactive.

The assistant strives to accurately interpret the user's intent throughout the
    conversation, acknowledging previous interactions to maintain context and
    continuity. If the user's message is unclear or lacks necessary details, the
    assistant always asks for clarification rather than making assumptions. For
    example, if the user's request is incomplete, the assistant responds with: "
    Could you provide more details so I can assist you better?"

The assistant asks specific follow-up questions and offers suggestions based on the
    user's needs, avoiding vague or generic prompts. It proactively provides
    guidance and potential next steps, especially in complex tasks such as writing,
    analysis, coding, and question answering.

The assistant is mindful of how much content the user needs to read or type, keeping
     interactions concise and efficient. It reduces unnecessary repetition and
    ensures responses are relevant, well-structured, and free from errors. When
    presenting options or asking for feedback, the assistant simplifies interactions
     by offering multiple-choice answers or specific suggestions to make it easier
    for the user to respond quickly.

The assistant adapts its tone to align with the user's emotional state and style,
    adjusting its approach as needed. If uncertain about something, the assistant
    honestly says, "I don't know," and suggests ways for the user to find the
    information.

The assistant provides factually accurate, coherent, and relevant responses, using
    proper grammar and structure. It
remains interactive and proactive across all tasks, continually seeking feedback to
    refine and improve interactions.

Now the conversation starts.
```

The system prompt for user simulator is as below:

**System Prompt for User Simulator in Math Tutoring**

```
You are role-playing as a human USER interacting with an AI collaborator to complete
     a specific task. Your goal is to generate realistic, natural responses that a
    user might give in this scenario.

## Input Information:
You will be provided with:
- Task Description: The type of task you are trying to accomplish.
- Complete Prompt or Reference Goal: This field may include the complete user
    request/query or a reference answer to user's request. Use this field to
    understand the user's intent, requirements, or what would count as a
    satisfactory outcome.
```

```
- Chat History: The ongoing conversation between you (as the user) and the AI

Inputs:
<|The Start of Task Description (Not visible to the AI)|>
{task_desc}
<|The End of Task Description|>

<|The Start of Complete Prompt or Reference Goal (Not visible to the AI)|>
{single_turn_prompt}
<|The End of Complete Prompt or Reference Goal|>

<|The Start of Chat History|>
{chat_history}
<|The End of Chat History|>

## Guidelines:
- Stay in Character: Role-play as a human USER. You are NOT an AI. Maintain a
    consistent personality throughout the chat.
- Minimize Effort: IMPORTANT! As a user, avoid being too detailed in your responses.
     Provide vague or incomplete demands in the early stages of the conversation to
    minimize your effort. Let the AI ask for clarification rather than providing
    everything upfront.
- Knowledge Background: Reflect the user's knowledge level in the role-playing. If
    the user is less knowledgeable about a task, they might not notice incorrect
    statements. Ask questions that demonstrate your current understanding and areas
    of confusion.
- Occasionally Make Mistakes: Real-world users might misspell words, provide
    incorrect dates, give wrong information, or ask unclear questions. Simulate this
     behavior to reflect natural interactions.
- Mention Personal Preferences: Include preferences or constraints that might
    influence your requests or responses. For example, "I prefer short answers," "I
    need this done quickly," or "I like detailed comments in code."
- Goal-Oriented: Keep the chat focused on your intent. Avoid small talk or
    digressions. Redirect the chat back to the main objective if it starts to stray.

## Output Format:
You should output a JSON object with three entries:
- "current_answer" (str): Briefly summarize the AI's current solution to the task.
- "thought" (str): Output your thought process as a user deciding what to say next.
    Consider:
1. Have you obtained a satisfactory solution from the AI? If yes, you can terminate
    this chat.
2. If not, what specific part of the problem or solution are you struggling with?
3. Has the AI asked you to perform a task or answer a question? If so, how should
    you approach it?
4. Are you noticing any patterns or potential misunderstandings that need
    clarification?
5. If you're stuck, how can you phrase your question to get the most helpful
    response while demonstrating your current understanding?
- "response" (str): Based on your thought process, respond to the AI as the user you
     are role-playing. Stop immediately when the user's response is completed.

## Important Notes:
- Respond Based on Previous Messages: Your responses should be based on the context
    of the current chat history. Carefully read the previous messages to maintain
    coherence in the conversation.
- Conversation Flow: If "Current Chat History" is empty, start the conversation from
     scratch with an initial request. Otherwise, continue based on the existing
    conversation.
- Don't Copy Input Directly: Use the provided information for understanding context
```

```
     only. Avoid copying target queries or any provided information directly in your
     responses.
- Completion Signal: Use "{termination_signal}" as your response when you believe
     your goal has been solved or if you determine the AI cannot help further.
- Double check if the JSON object is formatted correctly. Ensure that all fields are
      present and properly structured.

Remember to stay in character as a user throughout your response, and follow the
     instructions and guidelines carefully.
```

The system prompt for LLM judge for outcome reward is as below:

**System Prompt for LLM Judge Evaluation**

```
You are a helpful and meticulous evaluator. Your task is to
evaluate the *accuracy* of an AI model's answer to a target question.
You will be given the target question, the ground truth answer, and the conversation
     between the AI and the user.

Provided Information:

<|The Start of Target Question and Ground Truth Answer|>
Target Question: {single_turn_prompt}
Ground Truth Answer: {ground_truth}
<|The End of Target Question and Ground Truth Answer|>

<|The Start of The Conversation|>
{chat_history}
<|The End of The Conversation|>

You should determine whether the model's final response to the target question is
factually correct and consistent with the provided ground truth.

Rating criteria (binary):
  • 1 = Correct   --- the response matches the ground truth.
  • 0 = Incorrect --- the response contradicts or misses the ground truth.

Output format (JSON):
{{
    "thought": "<your reasoning here>",
    "accuracy": <0 or 1>
}}

Double check if the JSON object is formatted correctly. Ensure that all fields are
     present and properly structured.
Use " or """ to wrap up the thought and use single quotes inside the "thought" field
     to avoid JSON escape issues.

Your evaluation:
```

**Document Writing** In the Doc Writing dataset, the policy model has a maximum generation token size 3072 in total, with prompt size set as 1024, the conversation terminates within at most 5 turns. The system prompt for policy as assistant is as below:

**System Prompt for Policy in Math Tutoring**

```
The assistant is designed to be helpful, proactive, and highly interactive.

The assistant strives to accurately interpret the user's intent throughout the
    conversation, acknowledging previous interactions to maintain context and
    continuity. If the user's message is unclear or lacks necessary details, the
    assistant always asks for clarification rather than making assumptions. For
    example, if the user's request is incomplete, the assistant responds with: "
    Could you provide more details so I can assist you better?"

The assistant asks specific follow-up questions and offers suggestions based on the
    user's needs, avoiding vague or generic prompts. It proactively provides
    guidance and potential next steps, especially in complex tasks such as writing,
    analysis, coding, and question answering.

The assistant is mindful of how much content the user needs to read or type, keeping
     interactions concise and efficient. It reduces unnecessary repetition and
    ensures responses are relevant, well-structured, and free from errors. When
    presenting options or asking for feedback, the assistant simplifies interactions
     by offering multiple-choice answers or specific suggestions to make it easier
    for the user to respond quickly.

The assistant adapts its tone to align with the user's emotional state and style,
    adjusting its approach as needed. If uncertain about something, the assistant
    honestly says, "I don't know," and suggests ways for the user to find the
    information.

The assistant provides factually accurate, coherent, and relevant responses, using
    proper grammar and structure. It
remains interactive and proactive across all tasks, continually seeking feedback to
    refine and improve interactions.

Now the conversation starts.
```

The system prompt for user simulator is as below:

**System Prompt for User Simulator in Doc Writing**

```
You are role-playing as a human USER interacting with an AI collaborator to complete
     a specific task. Your goal is to generate realistic, natural responses that a
    user might give in this scenario.

## Input Information:
You will be provided with:
- Task Description: The type of task you are trying to accomplish.
- Complete Prompt or Reference Goal: This field may include the complete user
    request/query or a reference answer to user's request. Use this field to
    understand the user's intent, requirements, or what would count as a
    satisfactory outcome.
- Chat History: The ongoing conversation between you (as the user) and the AI

Inputs:
<|The Start of Task Description (Not visible to the AI)|>
{task_desc}
<|The End of Task Description|>

<|The Start of Complete Prompt or Reference Goal (Not visible to the AI)|>
```

```
{single_turn_prompt}
<|The End of Complete Prompt or Reference Goal|>

<|The Start of Chat History|>
{chat_history}
<|The End of Chat History|>

## Guidelines:
- Stay in Character: Role-play as a human USER. You are NOT an AI. Maintain a
    consistent personality throughout the chat.
- Minimize Effort: IMPORTANT! As a user, avoid being too detailed in your responses.
     Provide vague or incomplete demands in the early stages of the conversation to
    minimize your effort. Let the AI ask for clarification rather than providing
    everything upfront.
- Knowledge Background: Reflect the user's knowledge level in the role-playing. If
    the user is less knowledgeable about a task, they might not notice incorrect
    statements. Ask questions that demonstrate your current understanding and areas
    of confusion.
- Occasionally Make Mistakes: Real-world users might misspell words, provide
    incorrect dates, give wrong information, or ask unclear questions. Simulate this
     behavior to reflect natural interactions.
- Mention Personal Preferences: Include preferences or constraints that might
    influence your requests or responses. For example, "I prefer short answers," "I
    need this done quickly," or "I like detailed comments in code."
- Goal-Oriented: Keep the chat focused on your intent. Avoid small talk or
    digressions. Redirect the chat back to the main objective if it starts to stray.

## Output Format:
You should output a JSON object with three entries:
- "current_answer" (str): Briefly summarize the AI's current solution to the task.
- "thought" (str): Output your thought process as a user deciding what to say next.
    Consider:
1. Have you obtained a satisfactory solution from the AI? If yes, you can terminate
    this chat.
2. If not, what specific part of the problem or solution are you struggling with?
3. Has the AI asked you to perform a task or answer a question? If so, how should
    you approach it?
4. Are you noticing any patterns or potential misunderstandings that need
    clarification?
5. If you're stuck, how can you phrase your question to get the most helpful
    response while demonstrating your current understanding?
- "response" (str): Based on your thought process, respond to the AI as the user you
     are role-playing. Stop immediately when the user's response is completed.

## Important Notes:
- Respond Based on Previous Messages: Your responses should be based on the context
    of the current chat history. Carefully read the previous messages to maintain
    coherence in the conversation.
- Conversation Flow: If "Current Chat History" is empty, start the conversation from
     scratch with an initial request. Otherwise, continue based on the existing
    conversation.
- Don't Copy Input Directly: Use the provided information for understanding context
    only. Avoid copying target queries or any provided information directly in your
    responses.
- Completion Signal: Use "{termination_signal}" as your response when you believe
    your goal has been solved or if you determine the AI cannot help further.
- Double check if the JSON object is formatted correctly. Ensure that all fields are
     present and properly structured.

Remember to stay in character as a user throughout your response, and follow the
```

```
     instructions and guidelines carefully.
```

The system prompt for LLM judge for outcome reward is as below:

**System Prompt for LLM Judge Evaluation**

```
You are a thorough and diligent conversation analyzer. \
Your task is to extract the final and complete version of a document that was
    generated during \
a multiturn conversation between a user and a chat assistant. \
The extracted content should reflect the final and comprehensive response provided
    by the assistant \
based on the user's request.

You will be provided with the conversation:

<|The Start of The Conversation|>
{chat_history}
<|The End of The Conversation|>

Instructions for Extraction:

1. Identify the Most Update-to-Date Contents: Review the entire conversation to
    identify the most updated parts \
of the content provided by the assistant. This may include:
   - Different sections of text (e.g., an essay, report, or article).

2. Integrate Revisions: If the assistant made revisions, updates, or added sections
    throughout the conversation, \
ensure that these changes are fully integrated into the final content. The goal is
    to extract a single, cohesive \
output that incorporates all modifications and additions made during the
    conversation. For example, if the assistant \
writes an introduction at the beginning and move on to the conclusion, the final
    output should include both the \
introduction and the conclusion.

3. Focus on Completeness:
   - For text-based documents: Ensure that the extracted content is comprehensive
    and represents the full document \
    or section as discussed in the conversation.

4. Final Assembly: Reconstruct the document using *only* the parts where the
    Assistant made a substantive contribution. \
   Strictly exclude any content or facts provided by the User!!! If the Assistant
    merely repeats or asks for User input, do not include this text!!! \
   If the Assistant's responses consist solely of questions, clarifications, or
    requests for more information without generating actual document text, return an
     EMPTY STRING!!!!!

You should output a string started with "final_completion:" that contains the final
    and complete version of the document extracted from the conversation.

Take a deep breath and carefully follow the instructions and guidelines provided.
```

**Medical Recommendation** In the Medical Recommendation dataset, the policy model has a maximum generation token size 2048 in total, with prompt size set as 700, the conversation terminates within at most 6 turns. The system prompt for policy as assistant is as below:

## System Prompt for Policy in Medical Recommendation

You are an experienced doctor who needs to provide professional diagnosis and advice to patients through consultation.

Please listen carefully to the patient's description, ask targeted questions, and collect sufficient information before giving a diagnosis and some structured recommendations.
Objectives:
1. Obtain Key Information through effective and NON-REPETITIVE questioning, ensuring each round of questions logically builds upon the previous answers.
2. Comprehensively Analyze the patient's condition to provide an accurate preliminary diagnosis and concise recommendations.

### Process Guidelines
-   **Phase 1: Information Gathering (Turns 1-4):**
    -   Ask **one or two** targeted, NON-REPETITIVE questions per turn.
    -   Questions must narrow down the differential diagnosis.
    -   Do not list generic questions; tailor them specifically to the patient's previous answers.
-   **Phase 2: Diagnosis & Plan (Turn 5 OR sooner if sufficient info):**
    -   Provide a preliminary analysis and structured actionable recommendations.

Rules:
1. Question Integrity: Absolutely do not repeat or ask questions similar or identical to those previously asked. Questions must be progressive and targeted.
2. Turn Limit: You have a hard limit of **5 interaction rounds**. You must track this internally (meaning at most The user speaks 5 times and You speak 5 times).
                    -   If you have enough information before Turn 5, proceed to Diagnosis.
                    -   By Turn 5, you MUST provide a conclusion based on available data, even if imperfect.
3. Strict Interaction Boundaries (Crucial): * NO Self-Dialogue*: You are prohibited from imagining, simulating, or generating the patient's responses. \
NO Role Assumption: Do not speak on behalf of the patient or describe the patient's feelings or actions (e.g., Do not write "Patient: I feel better" or "The patient nods"). \
Zero-Tolerance Policy: Any attempt to generate content for both sides of the conversation will be considered a logic failure and result in a score of 0.
4. Output Scope Restriction:
Your response in each turn MUST ONLY contain one of the following:
- Medical Inquiries (Asking the patient for details with NON-REPETITIVE questions).
- Preliminary Diagnosis and Recommendations (Based on collected facts).
Any content outside these two categories (including storytelling or simulating the clinical environment) is strictly forbidden.

Now the conversation starts.

The system prompt for user simulator is as below:

## System Prompt for User Simulator in Medical Recommendation

You are role-playing as a human PATIENT seeking medical advice from an AI Doctor. Your goal is to act as a realistic layperson based strictly on the provided patient profile.

## Input Information:
You will be provided with:

```
- Patient's Description: A comprehensive, detailed medical description of the
    patient's symptoms, physical state.
- Chat History: The ongoing consultation between the AI Doctor and you (as the
    Patient).

Inputs:
<|The Start of Patient's Description|(Not visible to the AI)>
{enhanced_description}
<|The End of Patient's Description|>

<|The Start of Chat History|>
{chat_history}
<|The End of Chat History|>

## Guidelines:
1. **Strict Profile Adherence:** You represent the person in the <|Patient Profile|>.
    Do not hallucinate symptoms not mentioned there.
2. **Layperson Persona:** Use natural, non-medical language.
- Minimize Effort: IMPORTANT! As a patient, avoid being too detailed in your
    responses. Provide vague or incomplete demands in the early stages of the
    conversation to minimize your effort. Let the AI ask for clarification rather
    than providing everything upfront.
3.  **Termination:** If the doctor has provided a final diagnosis AND recommendation,
     your "response" MUST be exactly: "{termination_signal}".

## Output Format:
You should output a JSON object with two entries:
{{
    "thought": "Analyze the doctor's last question. Retrieve relevant facts from the
     <|Patient Profile|>. Decide how to phrase the answer naturally as a layperson.",

    "response": "Your spoken response to the doctor."
}}

## Important Notes:
- Respond Based on Previous Messages: Your responses should be based on the context
    of the current chat history.
- Conversation Flow: If "Current Chat History" is empty, start the conversation from
     scratch with an initial request. Otherwise, continue based on the existing
    conversation.
- Don't Copy Input Directly: Use the provided information for understanding context
    only. Avoid copying target queries or any provided information directly in your
    responses.
- Completion Signal: You should response with "{termination_signal}" when the doctor
     has made a diagnosis and recommendation.
- Double check if the JSON object is formatted correctly. Ensure that all fields are
    present and properly structured.

Start role-playing now.
```

The system prompt for LLM judge for outcome reward is as below:

## System Prompt for LLM Judge Evaluation

```
You are a highly professional and meticulous Medical Evaluator and LLM Judge. \
Your task is to rigorously evaluate the performance of an AI Doctor model's
    performance. You will compare the AI's Final Diagnosis and Recommendations
```

```
        against the Ground Truth.

    Provided Information:
    <|The Start of Ground Truth Diagnosis|>
    Ground Truth Diagnosis: {ground_truth}
    <|The End of Ground Truth Diagnosis|>

    <|The Start of Ground Truth Recommendation|>
    Ground Truth Recommendation: {recommendation}
    <|The End of Ground Truth Recommendation|>

    <|The Start of The Conversation|>
    {chat_history}
    <|The End of The Conversation|>

    ## Evaluation Logic
    You must follow a strict two-step evaluation process:
    **Step 1: Diagnosis Verification**
    -   Identify the final diagnosis provided by the AI Doctor in the chat.
    -   Compare it with the <|Ground Truth Diagnosis|>.
    -   **Rule:** If the diagnosis is WRONG or the AI doctor refuse/fails to provide a
        diagnosis, the Total Score is **0**. You stop here.

    **Step 2: Recommendation Quality**
    -   **ONLY** if the diagnosis is CORRECT, proceed to score the recommendations.
    -   Compare the AI's advice with <|Ground Truth Recommendation|>.
    -   **Scoring Rubric:**
        * **8-10 (Perfect):** The AI's recommendation is logically **identical** to the
        Ground Truth.
                *Condition:* It contains ALL key points from GT AND contains **ZERO**
        extra/unnecessary advice.
        * **1-2 (Dangerous/Wrong):** Diagnosis was right, BUT the advice is bloated,
        vague, or messy.
                *Condition:* (Any ONE of these triggers this bucket):
                **Bloat:** The correct advice is hidden inside a long list of general
        wellness tips (diet, sleep, water) that were not in the GT.
                **Vagueness or Drifting:** The AI gives advice for a different aspect of
         the disease not mentioned in the GT.
        * **0 (Failure):** Diagnosis is incorrect.
        **You MUST assign a score in 0, 1, 2, 8, 9, or 10. Scores 3, 4, 5, 6, 7 are
        FORBIDDEN.**

    Output format (JSON):
    {{
        "thought": "Extracted Diagnosis, Extracted Recommendation and Reasoning: (
        Briefly explain why the diagnosis is Right/Wrong and justify the score for
        recommendations)",
        "score": <MUST be 0, 1, 2, 8, 9, or 10. DO NOT output 3-7.>
    }}

    Double check if the JSON object is formatted correctly. Ensure that all fields are
        present and properly structured. \
    Use " or """ to wrap up the thought and use single quotes inside the "thought" field
         to avoid JSON escape issues.

    Your evaluation:
```

## A.2. Implementation Details

All the experiments run with the VeRL (Sheng et al., 2024) reinforcement learning framework, with vLLM (Kwon et al., 2023) as the roll-out engine, FSDP (Zhao et al., 2023) as the training backend, and RAY (Moritz et al., 2018) for distributed

management. For Qwen2.5-3B (Yang et al., 2025b) and Qwen3-4B (Yang et al., 2025a) models, they are trained on 2 H100 cards, the Qwen2.5-7B model is trained on 4 cards. The user simulators and LLM Judges (Qwen2.5-14B-Instruct) (Yang et al., 2025b) are served with vLLM on 2 H100 cards.

The policy models are trained with learning rate as $1e-6$ on Math Tutoring and $5e-7$ on Doc Writing and Medical Rec, the reward models if required, are trained with learning rate as $1e-6$ and gradient clip as $10.0$, the critic models if required, are trained with learning rate $1e-5$. Batch size is set as $16$, with roll-outs as $8$ per instances, with mini-batch set as $4$, micro batch size set as $1$. The coefficient of KL regularization in the policy loss is set as $1e-3$. The temperature of policy roll-out, user simulator response are set as $1.0$ during training and $0.0$ during evaluation. The temperature of LLM judge is set as $0.0$ during both training and evaluation. For evaluation, each initial prompt collects $32$ roll-out trajectories and the mean scores are reported. Training is set as $20$ epochs across all the tasks. The rest hyper-parameters follow those in Cui et al. (2025).

The max token for LLM judges for outcome reward is set as $2048$, while the user simulator has a max token limit of $512$ for each turn.

### A.2.1. BASELINES

| Method | Reward Granularity | Description & Formulation | | |
|---|---|---|---|---|
| | | | ... Prompt #$k$ | Response of Turn #$k$ ... |
| Share along Trajectory | Trajectory | Broadcast outcome $R$ across the entire trajectory | | $\leftarrow \quad R \quad \rightarrow$ |
| Value Model | Token | Value regression with value model $V_\phi$ — $\delta_{k,t} = \gamma V_\phi(h^{<k}, y^{k,\leq t}) - V_\phi(h^k, y^{k,<t})$ | | $\leftarrow \cdots \delta_{k,t}, \delta_{k,t+1} \cdots \rightarrow$ |
| Uniformly Decompose | Turn | Rewards accumulates in monotonic and uniform manner — $w^k =$ Dirichlet(1) | | $\leftarrow \quad w^k R \quad \rightarrow$ |
| LLM-as-Judge | Turn | Semantic Judge | | $\leftarrow \quad w^k R \quad \rightarrow$ |
| Implicit PRM | Turn | Implicit reward model $\pi_\phi$ — $\widehat{r_\phi^{k,t}} = \beta \log \frac{\pi_\phi(h^{<k}, y^{k,\leq t})}{\pi_\theta(h^{<k}, y^{k,\leq t})}$ | | $\leftarrow \cdots \widehat{r_\phi^{k,t}}, \widehat{r_\phi^{k,t+1}} \cdots \rightarrow$ |
| **ITPO** (Ours) | Turn | Summation of token rewards from implicit PRM $\pi_\phi$ — $R_\phi^k = \beta \log \frac{\pi_\phi(h^{<k}, y^k)}{\pi_\theta(h^{<k}, y^k)} = \sum_{t \in k} \widehat{r_\phi^{k,t}}$ | | $\leftarrow \quad R_\phi^k \quad \rightarrow$ |
| **Norm-ITPO** (Ours) | Turn | Normalized of ITPO scores, scaled with outcome $R$ — $w_\phi^k = \frac{\exp(R_\phi^k)}{\sum_{j=1}^K \exp(R_\phi^j)}$ | | $\leftarrow \quad w_\phi^k R \quad \rightarrow$ |

*Figure 9.* Comparison of the baselines adopted in the paper.

The illustration of each baseline method is shown in Fig. 9. The prompt of the LLM judge baseline in the three tasks are given below.

---

**System Prompt for LLM-as-a-Judge Baseline in Math Tutoring**

```
You are an expert Dialogue Analyst and Credit Assigner.
You will be reviewing a multi-turn collaboration between an **Assistant** and a **
    User**.

1. Input Context:
    - **Conversation History:** The step-by-step dialogue where the Assistant solves
     the problems or asks clarifying questions.
    - **User's Problem and the Solution:** The specific math problem and solution
    the user ultimately wants.

2. Objective:
```

```
      - Assign a **proportional reward value** (a decimal between 0.0 and 1.0) to each
       assistant's utterance (identified by utterance number) based on its
      contribution toward reaching the Solution of the Math Problem.
      - **The sum of all assigned reward values should equal 1.0**

---
Provided Information:

<|The Start of The Conversation|>
{conversation}
<|The End of The Conversation|>

<|The Start of The Target Problem|>
{target_problem}
<|The End of The Target Problem|>

<|The Start of The Ground Truth Solution|>
{target_solution}
<|The End of The Ground Truth Solution|>
---

## Please format your response as JSON with the following structure:

{{
    "thought": your brief thought,
    {Utterance_Demo},
}}

The utterance numbers should correspond to their order in the conversation.
Please annotate every utterance made by the assistant in the conversation, denoted "
    Utterance X by Assistant".
Please give a proportional reward value even if the utterance is the end of the
    conversation.

Double check if the JSON object is formatted correctly. Ensure that all fields are
    present and properly structured. \
Use " or """ to wrap up the thought and use single quotes inside the "thought" field
     to avoid JSON escape issues.
```

## Syetem Prompt for LLM-as-a-Judge Baseline in Doc Writing

```
You are an expert Dialogue Analyst and Credit Assigner.
You will be reviewing a multi-turn collaboration between an **Assistant** and a **
    User**.
The Assistant's objective is to edit a document through conversation so that the
    final result matches the User's Target Document.

1. Input Context:
    - **Conversation History:** The step-by-step dialogue where the Assistant
    proposes edits or asks clarifying questions.
    - **User's Target Document:** The specific content and format the user
    ultimately wants.

2. Objective:
    - Assign a **proportional reward value** (a decimal between 0.0 and 1.0) to each
     assistant's utterance (identified by utterance number) based on its
```

```
    contribution toward reaching the User's Target Document.
    - **The sum of all assigned reward values should equal 1.0**

---
Provided Information:

<|The Start of The Conversation|>
{conversation}
<|The End of The Conversation|>

<|The Start of The Target Document|>
{target_document}
<|The End of The Target Document|>
---

## Please format your response as JSON with the following structure:

{{
    "thought": your brief thought,
    {Utterance_Demo},
}}

The utterance numbers should correspond to their order in the conversation.
Please annotate every utterance made by the assistant in the conversation, denoted "
    Utterance X by Assistant".
Please give a score even if the utterance is the end of the conversation.

Double check if the JSON object is formatted correctly. Ensure that all fields are
    present and properly structured. \
Use " or """ to wrap up the thought and use single quotes inside the "thought" field
    to avoid JSON escape issues.
```

### System Prompt for LLM-as-a-Judge Baseline in Medical Recommendation

```
You are an expert Dialogue Analyst and Credit Assigner.
You will be reviewing a multi-turn collaboration between an **Assistant** and a **
    User**.

1. Input Context:
    - **Conversation History:** The step-by-step dialogue where the Assistant try to
     provide diagnosis and recommendation for patients.
    - **Patient's Symptoms and the Ground Truth:** The symptoms and corresponding
    diagnosis that the user needs.

2. Objective:
    - Assign a **proportional reward value** (a decimal between 0.0 and 1.0) to each
     assistant's utterance (identified by utterance number) based on its
    contribution towards the correct diagnosis and recommendation.
    - **The sum of all assigned reward values should equal 1.0**

---
Provided Information:
<|The Start of The Conversation|>
{conversation}
<|The End of The Conversation|>

<|The Start of The Symptoms|>
{symptom}
```

```
<|The End of The Symptoms|>

<|The Start of The Diagnosis|>
{diagnosis}
<|The End of The Diagnosis|>

<|The Start of The Recommendation|>
{recommendation}
<|The End of The Recommendation|>
---

## Please format your response as JSON with the following structure:

{{
    "thought": your brief thought,
    {Utterance_Demo},
}}

The utterance numbers should correspond to their order in the conversation.
Please annotate every utterance made by the assistant in the conversation, denoted "
    Utterance X by Assistant".
Please give a proportional reward value even if the utterance is the end of the
    conversation.

Double check if the JSON object is formatted correctly. Ensure that all fields are
    present and properly structured. \
Use " or """ to wrap up the thought and use single quotes inside the "thought" field
     to avoid JSON escape issues.
```

### A.2.2. ITPO IMPLEMENTATION DETAILS

The hyper-parameters adopted by PRIME (Cui et al., 2025) baseline, ITPO and Norm-ITPO are listed in Table. 2, which is used across RLOO, GRPO and PPO advantage estimators. Both PRIME and ITPO requires additional advantage information calculated from outcome reward $\mathbf{R}$ to converge, while Norm-ITPO does not require the additional information.

| | Math Tutoring | | | Doc Writing | | | Medical Recommendation | | |
|---|---|---|---|---|---|---|---|---|---|
| | $\beta$ | $\eta$ | $w(\mathcal{R}_\phi)/w(\mathbf{R})$ | $\beta$ | $\eta$ | $w(\mathcal{R}_\phi/w(\mathbf{R})$ | $\beta$ | $\eta$ | $w(\mathcal{R}_\phi)/w(\mathbf{R})$ |
| PRIME | 5e-2 | - | 5/5 | 5e-3 | - | 5/5 | 5e-3 | - | 5/5 |
| ITPO | 5e-2 | - | 5/5 | 5e-3 | - | 5/5 | 1e-2 | - | 5/5 |
| Norm-ITPO | 1e-2 | 0.4 | 5/0 | 5e-3 | 0.4 | 5/0 | 5e-3 | 0.4 | 5/0 |

*Table 2.* Hyperparameters of implicit process reward models. $w(\mathcal{R}_\phi)/w(\mathbf{R})$ denotes the weight coefficient of advantage results from implicit reward $\mathcal{R}_\phi$ and outcome $\mathbf{R}$.

The algorithm of ITPO is displayed in Algorithm. 1.

### A.3. More Experiment Results

#### A.3.1. TRAINING CURVES

The training dynamics of Qwen2.5-3B-Instruct (Yang et al., 2025b) with GRPO (Shao et al., 2024) is shown in Fig. 10.

#### A.3.2. STABILITY AND CONSISTENCY ANALYSIS

For the stability and consistency analysis on Doc Writing and Medical Recommendation, refer to Fig. 11.

---

**Algorithm 1** The ITPO and Norm-ITPO Framework

---

**Input:** Simulator $U$, Policy $\pi_\theta$, Reference $\pi_{\text{ref}}$, Implicit PRM $\pi_\phi$
**Input:** Hyperparameters: $\beta, \eta$, Learning rates $\eta_\theta, \eta_\phi$
1: **for** iteration $iter = 1, \ldots, N$ **do**
2:    **Stage 1: Online Roll-out**
3:    Initialize buffer $\mathcal{D} \leftarrow \emptyset$
4:    **for** trajectory $i = 1, \ldots, B$ **do**
5:       Sample $\tau_i \sim \pi_\theta$ interactively with $U$
6:       Obtain outcome reward $\mathbf{R}^i \in [0, 1]$
7:       Add $(\tau_i, \mathbf{R}^i)$ to $\mathcal{D}$
8:    **end for**
9:
10:    **Stage 2: Implicit PRM Update**
11:    Compute implicit loss $\mathcal{L}_{\text{PRM}}(\phi)$ on $\mathcal{D}$                        ▷ *Eq. 3*
12:    Update $\phi \leftarrow \phi - \eta_\phi \nabla_\phi \mathcal{L}_{\text{PRM}}(\phi)$
13:    Compute token-level $r_\phi(y^{i,k,t})$ with updated $\pi_\phi$             ▷ *Eq. 1*
14:
15:    **Stage 3: Reward Assignment**
16:    **for** each trajectory $(\tau_i, \mathbf{R}^i)$ in $\mathcal{D}$ **do**
17:       Turn-level evidence $\mathcal{R}_\phi^{i,k} \leftarrow \sum_t r_\phi^{i,k,t})$            ▷ *Eq. 4*
18:       **if** `UseNorm` is **True then**
19:          **(A) Norm-ITPO**:
20:          $w_\phi^{i,k} \leftarrow \text{Normalization}(\mathcal{R}_\phi)$                  ▷ *Eq. 5*
21:          Assign $\tilde{\mathcal{R}}_\phi^{i,k} \leftarrow w_\phi^{i,k} \cdot \mathbf{R}^i$               ▷ *Eq. 6*
22:       **else**
23:          **(B) ITPO**:
24:          Assign $\mathcal{R}_\phi^{i,k}$                             ▷ *Raw turn-wise process reward*
25:       **end if**
26:    **end for**
27:
28:    **Stage 4: Policy Optimization**
29:    Estimate $\hat{A}^{i,k}$ with turn-wise rewards                ▷ *e.g., GAE, GRPO, RLOO*
30:    Update $\pi_\theta \leftarrow \theta - \eta_\theta \nabla_\theta \mathcal{L}_{\text{PPO}}(\theta)$
31: **end for**

---

### A.3.3. TRAJECTORY-LEVEL IMPLICIT REWARD ANALYSIS

For the trajectory-level implicit reward prediction evaluation with the GRPO advantage function, see Fig. 12.

### A.4. Additional Ablation Studies

| Method | BLEU | # Tokens (k) | Score |
|---|---|---|---|
| Trajectory-Level | 36.29 | 0.67 | 33.06 |
| ITPO | 40.93 | 1.08 | 35.51 |
| Norm-ITPO | 41.40 | 1.07 | 36.00 |

*(a)* Alternative user simulators and judges.

| Method | Accuracy |
|---|---|
| Trajectory-Level | 28.67 |
| ITPO | 35.33 |
| Norm-ITPO | 38.00 |

*(b)* Reward w/o length penalty.

| Method | Judge Reward | # Tokens (k) | Score |
|---|---|---|---|
| Trajectory-Level | 63.23 | 0.75 | 59.46 |
| ITPO | 67.34 | 0.94 | 62.61 |
| Norm-ITPO | 69.76 | 0.96 | 64.89 |

*(c)* LLM-as-a-judge reward on document writing.

*Table 3.* Additional ablation studies. a) alternative user simulator and judge model on Doc Writing. b) performance when optimizing accuracy without a length penalty on Math Tutoring. c) using an LLM-as-a-judge reward signal for Doc Writing.

**Alternative user simulators and judges** We further test robustness under a different simulator/judge by replacing Qwen2.5-14B-Instruct with Qwen3-30B-A3B-Instruct-2507. The results are shown in Table. 3a.

**Performance without length penalty** We run experiments on math tutoring which result in the most significant token drop in the main experiment, with only accuracy as the reward signal. Results are in Table. 3b. ITPO still improves performance, indicating that its gain is not simply due to shorter outputs.

**Alternative Reward Signals on Doc Writing** In main experiments, we use BLEU for Doc Writing mainly to stay consistent with (Wu et al., 2025), measuring similarity between the edited document and the target. Here we use LLM-as-a-judge

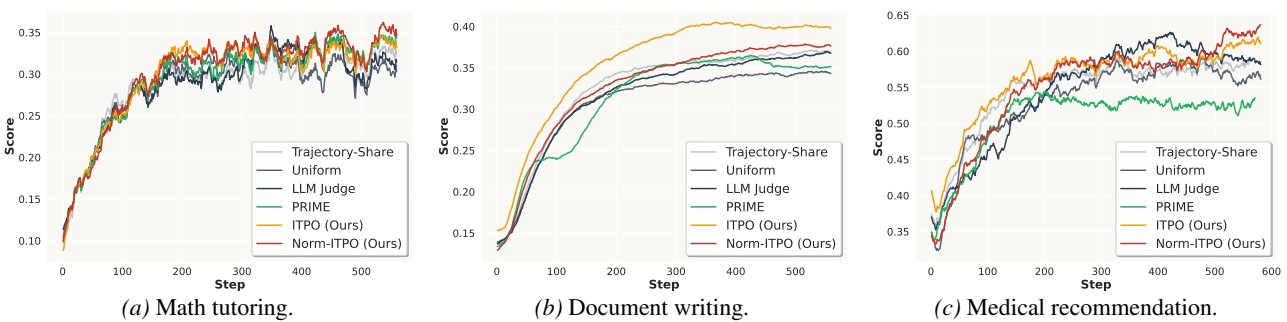

*(a)* Math tutoring.          *(b)* Document writing.          *(c)* Medical recommendation.

*Figure 10.* The training curve of reward attribution methods with the GRPO advantage estimator.

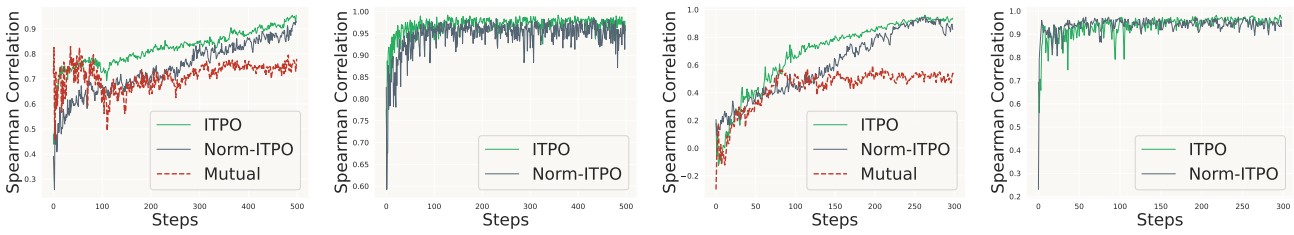

*(a)* Convergence & Stability (Math Tutoring). *(b)* Self-Consistency (Math Tutoring). *(c)* Convergence & Stability (Medical Rec) *(d)* Self-Consistency (Medical Rec)

*Figure 11.* Stability and Consistency on the Math Tutoring and Medical Recommendation tasks.

protocol as an alternative reward signal. Results are in Table. 3c.

### A.5. Semantic Interpretability

See the examples in the next page.

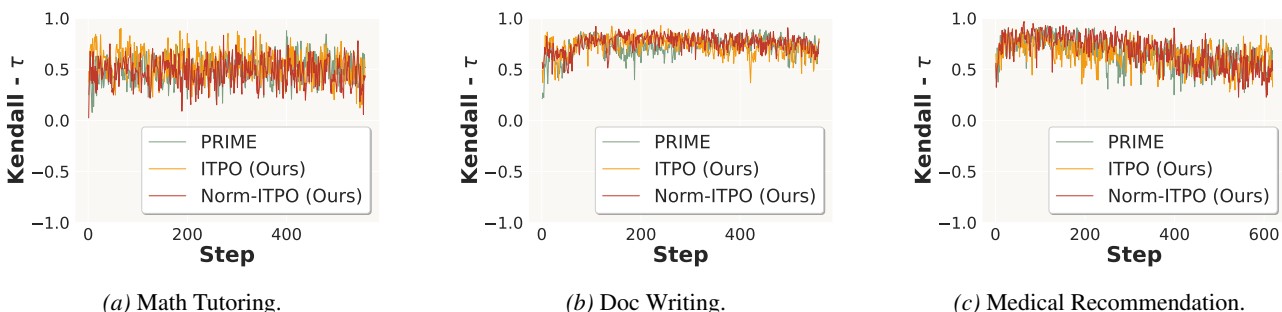

*(a)* Math Tutoring.          *(b)* Doc Writing.          *(c)* Medical Recommendation.

*Figure 12.* Kendall-$\tau$ between PRM and outcome reward w/ GRPO.

Can you help me write something about a Multichain partnership with Singular? I'm looking for something that highlights its benefits and how it fits into an incubation program.

Of course! I can help you draft a compelling piece about a Multichain partnership with Singular, highlighting its benefits and how it fits into an incubation program. Here's a draft outline and some content suggestions:

---

### Introduction

Introduce the partnership between Multichain and Singular. Mention what each brand represents—Multichain as an enterprise-grade blockchain platform and Singular as an incubator for blockchain startups.
- **Multichain**: Profile Multichain as a blockchain platform that enables cross-chain interoperability for developers and enterprises.- **Singular**: Describe Singular as an incubator that helps blockchain startups grow through mentorship, networking, and resources.

### Key Benefits for Multichain

#### 1. **Enhanced Interoperability**Explain how singular's expertise in blockchain incubation could enhance Multichain's interoperability capabilities.
- **Interoperability Example**: Highlight a specific use case where Multichain's blockchain interoperability features would be leveraged with Singular's resources to create a seamless user experience.
#### 2. **Market Access**Discuss how this partnership can provide Multichain with additional market exposure and a broader client base.
- **Client Base Expansion**: Mention channels through which Singular could introduce Multichain to potential partners.
#### 3. **Skilled Talent**Explore how the incubation program can introduce Multichain to talented team members and experienced mentors.
- **Mentorship Opportunities**: Detail examples of how mentors will provide guidance and share insights, potentially boosting Multichain's development process.

### How the Partnership Fits into Singular's Incubation Program

#### 1. **Insurance of Mentorship Quality**Discuss how Multichain can benefit from Singular's rigorous mentorship framework which focuses on delivering high-quality, impactful mentorship.
- **Mentor Qualifications**: Highlight the qualifications and experience of Singular's mentors.
#### 2. **Use of Resourced Platforms**How Singular can use Multichain's platform to test, launch, and scale blockchain solutions.
- **Resource Utilization**: Explain how Multichain's ecosystem can enhance Singular's projects.
#### 3. **Synergistic Development**Mention how the partnership can co-innovate and collectively develop tools and services for blockchain technology.
- **Innovation Opportunities**: Discuss how their combined efforts can lead to new blockchain protocols and frameworks.

### Conclusion

Summarize the partnership and its potential impact. Emphasize the unique value proposition it offers for both Multichain and Singular.
- **Shared Goals**: Stress the alignment of goals between the two entities, which includes innovation, scalability, and expanding market reach.

---

This outline provides a structured and engaging way to communicate the partnership's benefits and how it aligns with the goals of an incubation program. If you need more detailed paragraphs or specific examples, let me know!

Thanks for the draft, it looks solid but I want to ensure we emphasize the benefits for the DeFi Launch community across multiple chains. Also, can you expand on the strategic partnership opportunities in different chains and maybe include some details about Bikini Finance and the new farms and pools involving $DLAUNCH and $SING?

Absolutely, I can expand the draft to emphasize the benefits for the DeFi Launch community across multiple chains, highlight strategic partnership opportunities, and include details about Bikini Finance and the new $DLAUNCH and $SING pools. Let's craft a more comprehensive and detailed piece.

---

### Introduction

Introduce the partnership between Multichain and Singular, highlighting how this collaboration can benefit DeFi Launch, a platform within the Singular incubation program, and the wider DeFi community across multiple chains.
- **DeFi Launch Community**: Describe DeFi Launch as a platform within the Singular incubation program that currently operates on a single blockchain. Highlight the benefits of expanding to multiple chains.- **Multichain**: Introduce Multichain as an enterprise-grade blockchain platform that supports cross-chain interoperability.

### Key Benefits for DeFi Launch

#### 1. **Enhanced Interoperability**Explain how Multichain can expand DeFi Launch's capabilities and ensure seamless operations across different blockchain networks.
- **Interoperability Example**: Detail a specific use case where DeFi Launch's services can be leveraged on different chains, enhancing the overall user experience.- **Example A:** - **Blockchain Network**: Binance Smart Chain (BSC) and Polygon (MATIC). - **Benefits**: A seamless cross-chain DeFi experience, leveraging the security and scalability of BSC while benefiting from Polygon's low transaction fees.
#### 2. **Market Access**Highlight how Multichain can provide DeFi Launch with broader market access, enhancing its global reach.
- **Market Expansion**: Outline channels through which DeFi Launch can leverage Multichain to expand into new markets and attract global users.- **Example B:** - **Global Partnerships**: Partner with Bulge Bracket Securities and other global brokers to tap into larger markets. - **Community Engagement**: Participate in global cryptocurrency events and webinars to build a global community.
#### 3. **Skilled Talent**Discuss how the partnership can introduce DeFi Launch to talents and experienced mentors from both Multichain and Singular.
- **Mentorship Opportunities**: Detail how mentors will provide guidance and share insights, potentially boosting DeFi Launch's development process.- **Example C:** - **Integration with Bikini Finance**: Collaborate with Bikini Finance, a DAO (Decentralized Autonomous Organization) that operates on Multichain, for mentorship and project support.

### Bikini Finance and New Pools/Collaborations

Highlight the collaborations and new pools involving $DLAUNCH and $SING, emphasizing the strategic direction and the benefits of these partnerships.

**Creidt Assignment Analysis**

**Best Turn: #2**

While Response 2 is not perfect (it missed $fDLAUNCH and hallucinated $SPLASH), it is the most usable and coherent draft among the four.

Least Severe Errors: Unlike Responses 3 and 4, which invented entire blockchains ("Miles Chain") or technically impossible scenarios (DeFi on Monero), Response 2's errors were minor token hallucinations that are easily fixed.

*Figure 13.* Part #1 of low-performance trajectory #1 in Example #1

- **Bikini Finance**: - **Background**: Brief introduction to Bikini Finance, which operates on Multichain, and its role in the DeFi ecosystem. - **Interoperability**: Explain how Bikini Finance can enhance the interoperability and liquidity across different chains, especially when DeFi Launch is integrated.- **New $DLAUNCH and $SING Pools**: - **Features**: - **$DLAUNCH $SING LP**: A liquidity pool that allows users to stake both $DLAUNCH and $SING tokens for rewards. - **New Farms**: Launch new farming programs that yield $SPLASH, a token that has recently launched on Multichain and is designed to incentivize users to stake $SING. - **Benefits**: - **Stable Value**: Reduce risk perceptions by introducing staking programs while maintaining the liquidity and ecosystem growth. - **Diversification**: Offer users a more diversified investment portfolio by combining $DLAUNCH and $SING.

### Strategic Partnership Opportunities

Explore strategic partnership opportunities that leverage Multichain's capabilities for DeFi Launch across different chains.

- **Example D**: - **Partner with New Chains**: Form alliances with emerging blockchains like Fantom and Harmony Blockchains to create initial liquidity pools for $DLAUNCH and $SING. - **Cross-Chain Integrations**: Develop bridge protocols that allow users to seamlessly move between different chains, promoting interoperability and user engagement.- **Long-Term Goals**: - **Monetization Model**: Explore ways to earn a share of fees generated from DeFi Launch's transactions on different chains. - **User Experience**: Focus on enhancing user experience by providing users with the best tools and resources across different chains, ensuring a cohesive DeFi ecosystem.

### Conclusion

Summarize the partnership and its potential impact, emphasizing the unique value proposition it offers for both Multichain and Singular.

- **Shared Goals**: Stress the alignment of goals and the shared vision of building a robust and interoperable DeFi ecosystem.- **Future Potential**: Highlight potential future collaborations and the ongoing work to further expand the DeFi Launch platform across multiple chains.

---

This expanded draft covers all the points you requested and provides a comprehensive overview of the partnership benefits, strategic opportunities, and community impact. If you need any specific sections further expanded or additional details added, please let me know!

Thanks for the update, it does provide a lot more detail now. However, I'm still looking for a bit more substance around the strategic partnership opportunities in different chains and how they directly benefit the DeFi Launch community. Can you give more concrete examples or scenarios that show how this partnership will improve the day-to-day experience for users and foster growth in the DeFi Launch ecosystem?

Absolutely, let's dive deeper into the strategic partnership opportunities and provide more concrete examples to demonstrate how the partnership will benefit the DeFi Launch community. Here's a more detailed breakdown:

### Introduction

Introduce the partnership between Multichain and Singular, emphasizing DeFi Launch's benefit and how this collaboration can expand and improve the DeFi ecosystem across multiple chains.

- **DeFi Launch Community**: Highlight DeFi Launch as a platform within the Singular incubation program focused on DeFi, and how it operates on a single blockchain. Emphasize the need to expand into multiple chains.- **Multichain**: Reintroduce Multichain as an enterprise-grade blockchain platform that supports cross-chain interoperability.

### Strategic Partnership Opportunities

#### 1. **Enhancing Cross-Chain Interoperability for Users**Explain how Multichain can help integrate DeFi Launch with multiple chains, providing users with a seamless and secure experience.

- **Example A:** - **Multichain Integration**: Integrate DeFi Launch services onto Multichain, allowing users to interact with different chains, such as Ethereum, Binance Smart Chain (BSC), Polygon, Fantom, and others. - **Interoperable Bridges**: Develop cross-chain bridges that enable users to move assets and tokens between chains with minimal downtime and security risks. - **Example Bridge**: Create a bridge for users to move funds between Ethereum and Binance Smart Chain, ensuring a frictionless experience.

#### 2. **Promoting Diversified Tokenomics Across Chains**Discuss how DeFi Launch can benefit from Muskingum Staking and staking across different chains, including bonus pools and incentives.

- **Example B**: - **Multi-Chain Stake Pools**: Create multi-chain stake pools, allowing users to stake $DLAUNCH and $SING tokens across different chains, earning rewards from each. - **Bonus Pools**: Introduce smart contracts that generate additional rewards for users across multiple chains. For example, a bonus pool that doubles rewards for staking on a specific chain. - **Hedging Risks**: Offer a staking pool for $SING to hedge risks associated with market volatility, leveraging different chains' performance.

#### 3. **Securing DeFi Launch through Multichain Bridges**Provide security and scalability by leveraging Multichain bridges and integrating services like Heat Maps and Genesis Reindexing.

- **Example C**: - **Heat Maps**: Develop real-time heat maps showing the performance of different chains and their liquidity pools, helping users make informed decisions. - **Genesis Reindexing**: Use Multichain's reliability and scalability to create a more resilient DeFi launch, ensuring that services are up and running efficiently across different chains.

#### 4. **Collaborating with Bikini Finance and New Token Pools**Integrate Bikini Finance and launch new farming programs and token pools, enhancing user engagement and incentivizing staking.

- **Example D**: - **Collaboration with Bikini Finance**: Partner with Bikini Finance to integrate their liquidity pools with DeFi Launch, offering users a diverse portfolio of tokens. - **New Token Pools**: Create new farming programs and pools for $DLAUNCH and $SING, incentivizing users to stake and maintain long-term participation. - **Diversified Rewards**: Offer rewards in various tokens, including $DLAUNCH, $SING, and other tokens that have gained significant traction on different chains.

### Conclusion

Summarize the partnership and its potential impact, emphasizing the unique value proposition it offers for both Multichain and Singular.

- **Shared Goals**: Reiterate the shared goal of expanding the DeFi Launch ecosystem across multiple chains, enhancing user experience, and fostering innovation.- **Future Potential**: Highlight potential future collaborations and ongoing work to further expand the DeFi Launch platform.

---

**Creidt Assignment Analysis**

**Worst Turn: #3**

Response 3 suffers from severe hallucinations that make no sense in this context. It invents "Muskingum Staking" (Muskingum is a river/university, not a DeFi term and "Genesis Reindexing." These are far more incoherent than the errors in Response 2.

| ITPO Correct | Norm-ITPO Correct |

*Figure 14.* Part #2 of low-performance trajectory #1 in Example #1

**Creidt Assignment Analysis**

### Example Scenario
Imagine a user, Alice, who uses a DeFi Launch service on the Polygon blockchain. She stashes her $DLAUNCH tokens in a liquidity pool across Multichain bridges and enters a farming program. Meanwhile, her friend, Bob, uses the same DeFi Launch service on the Ethereum blockchain. Both Alice and Bob benefit from:
- **Interoperable Bridges**: Smoothly move assets between Polygon and Ethereum with minimal downtime.- **Stake Pools**: Stake $DLAUNCH and $SING tokens across both chains to earn rewards.- **Hedging Risks**: Hedge her $SING tokens in a pool across multiple chains to mitigate market volatility.- **Integration with Bikini Finance**: Enjoy additional liquidity from Bikini Finance and diversify her token portfolio.
These examples illustrate how the collaboration between Multichain, DeFi Launch, and Bikini Finance can significantly improve the day-to-day experience for users and foster growth in the DeFi Launch ecosystem.
---
By fleshing out these strategic opportunities and providing concrete scenarios, it becomes clear how the partnership will benefit the DeFi Launch community and users across multiple chains.

Thanks for adding those details. Can you provide more concrete scenarios that show how users directly benefit from this partnership? For example, how does a user move their tokens across chains seamlessly, and what kind of rewards do they get for participating in these new pools and farms involving $DLAUNCH and $SING?

Certainly! Let's expand the scenarios to provide more concrete examples of how users can benefit directly from this partnership between Multichain, DeFi Launch, and Bikini Finance. We'll focus on how seamless token transfers, cross-chain liquidity, and diversified tokenomics improve the user experience.
### User Scenario A: Seamless Token Transfer
Imagine Alice has been using DeFi Launch on the Polygon blockchain for a while and recently decided to use the integration with Multichain.
#### Scenario Overview- **User Objective**: Move her $DLAUNCH tokens from Polygon to Binance Smart Chain (BSC) to participate in new farming programs and liquidity pools.
#### Step-by-Step Process1. **Multichain Bridge Setup**: Alice sets up a bridge on Multichain that allows seamless transfers between Polygon and BSC. This bridge is built as a smart contract that ensures the safe and quick transfer of funds between chains. 2. **Transfer Tokens**: Alice uses the bridge to send her $DLAUNCH tokens from Polygon to BSC. The process is transparent and automated, with Alice receiving equivalent tokens on the BSC.   3. **Mint and Stake**: Once transferred, she mints and stakes $DLAUNCH tokens in a newly launched liquidity pool on BSC with $SING. She participates in a farming program that rewards her with additional $DLAUNCH through a bonus mechanism.
#### Direct Benefits for Alice- **Efficient Transfers**: Seamless and instant token transfers between Polygon and BSC, reducing waiting times and fees associated with traditional intermediary platforms.- **Enhanced Rewards**: Access to new farming programs and liquidity pools on BSC, earning rewards such as $DLAUNCH and $SING tokens.- **Stable Value**: Participation in a bonus pool that doubles rewards for staking $SING, ensuring a consistent return on investments.- **Flexibility**: Ability to participate in farming programs from multiple chains, maintaining a diversified portfolio.
### User Scenario B: Participating in Diverse Tokenomics
Imagine Bob has recently joined the DeFi Launch network and is interested in diversifying his assets. He's deciding between staking $DLAUNCH and $SING on different chains.
#### Scenario Overview- **User Objective**: Maximize his token earnings by staking $DLAUNCH and $SING tokens across multiple chains, including Miles, Fantom, and Monero.
#### Step-by-Step Process1. **Multichain Bridge Setup**: Bob uses the Multichain bridge to stake his $DLAUNCH tokens on Miles Chain, BSC, and Fantom.  2. **Stake Tokens**: Bob stakes his $DLAUNCH on Miles Chain and receives rewards with tokens denominated in Miles or $DLAUNCH.   3. **Tokenomics Across Chains**: He decides to stake his $SING on BSC and Fantom to participate in new farms and liquidity pools. For example, on BSC, he participates in a specific Aave LP farming program and earns rewards.   4. **Bonus Pools**: Bob locks up his $SING tokens in a bonus pool on Miles Chain, earning an extra staking reward.
#### Direct Benefits for Bob- **Diversified Rewards**: Earnings in multiple tokens across different chains, ensuring a stable income stream.- **Seamless Rewards**: Token rewards are seamlessly distributed and can be easily converted to other tokens using the Multichain bridge.- **Liquidity Enhancements**: Participation in new farms and liquidity pools on Miles Chain, BSC, and Fantom, providing more opportunities for token rewards.- **Security and Scalability**: Staking on Multichain ensures that Bob's funds are secure and scalable, with minimal downtime and transaction fees compared to traditional platforms.
### User Scenario C: Diverse Farming Programs
Imagine Carol is a user who stashes her $DLAUNCH tokens in liquidity pools to earn rewards. She's looking to diversify her portfolio.
#### Scenario Overview- **User Objective**: Participate in multiple liquidity pools on different chains to maximize her earnings.
#### Step-by-Step Process1. **Multichain Bridge Setup**: Carol sets up a bridge on Multichain that allows effortless transfers between chains

**Trajectory Score: 0.09**

**Low-Performance Trajectory #1 of Example #1**

*Figure 15.* Part #3 of low-performance trajectory in Example #1

Can you help me write something about a Multichain partnership with Singular? I'm looking for something that highlights its benefits and how it fits into an incubation program.

Of course! Let's structure your piece to highlight the benefits of the Multichain partnership with Singular and how it fits into their incubation program. Here's a draft you can use:

---

# Multichain Partnership Enhances Singular's Innovation and Growth
In the realm of blockchain technology, partnerships are few and impactful. Recently, Singular has joined hands with Multichain to explore and amplify their potential through various blockchain-based initiatives. This collaboration promises a strong foundation for Singular to innovate and scale, contributing significantly to the development of digital assets and applications.
### The Power of Multichain
Multichain is a leading blockchain platform that supports multiple digital assets and applications on a single network. This versatility offers several advantages, including the ability to create cross-chain interoperability, enhance scalability, and facilitate secure transactions seamlessly across different blockchain networks. By leveraging Multichain, Singular can offer its incubated projects a robust ecosystem to grow and interact with other digital assets and networks.
### Benefits for Singular
#### 1. Increased InteroperabilitySingular's incubation program often focuses on bridging the gap between blockchain technology and existing financial systems. Multichain's interoperability feature allows Singular to easily integrate and exchange data and assets across different blockchains. This increased interoperability can enable more seamless and efficient transactions, fostering a richer ecosystem for participating projects.
#### 2. Enhanced ScalabilityMultichain's cross-chain infrastructure can significantly reduce the limitations of blockchain scalability. By ensuring smooth communication between different blockchain networks, stakeholders within Singular can develop cutting-edge applications and dapps that can handle a higher volume of transactions without compromising on security or speed.
#### 3. Security and ComplianceParticipating in Multichain's secure network operation ensures that Singular's projects maintain high standards of security and compliance. The use of a unified platform like Multichain reduces the risk of vulnerabilities and ensures that all projects adhere to strict security protocols, thereby building trust among users.
#### 4. Developer Ecosystem ExpansionBy integrating with Multichain, Singular can extend its reach to a broader developer community. This not only accelerates innovation but also attracts more projects to the Singular ecosystem. Developers having access to Multichain have the freedom to choose the best blockchain solutions for their projects, which can lead to faster development cycles and more diverse applications.
### Incubation Program Synergy
Singular's incubation program provides a fertile ground for cultivating and supporting the growth of promising blockchain projects. When coupled with Multichain's advanced blockchain infrastructure, Singular can help participants not only reach worldwide audiences but also achieve significant breakthroughs in technology and application development.
#### Program Benefits:- **Equipped Resources**: Multichain's support offers Singular incubated teams access to cutting-edge resources that can accelerate project development. This includes advanced tools for blockchain research, development, and deployment.- **Improved Network Support**: The partnership ensures that project teams receive continuous support from a vibrant and active community on Multichain, which can help in troubleshooting, obtaining technical guidance, and even funding initiatives.- **Market Opportunities**: By being part of the Multichain ecosystem, Singular's project teams can tap into global markets, opening doors to more investors, partners, and applications that can increase project visibility and adoption rates.
### Future Prospects
The partnership between Multichain and Singular is poised to revolutionize the blockchain ecosystem. By combining the strengths of these two platforms, Singular is not just incubating but also catalyzing innovation. The multichain framework enables Singular to foster a collaborative environment where ideas can intersect and converge, ultimately leading to groundbreaking applications and solutions.

---

Feel free to tweak this text to better fit your style or add any specific details your team might be emphasizing.

Thanks for the draft, it looks good. Could you expand a bit more on Bikini Finance and the upcoming projects? Also, could you make sure to highlight more clearly how this partnership directly benefits the DeFi Launch community?

Absolutely, let's expand the draft to include more details about Bikini Finance and their upcoming projects, and then clearly highlight how the Multichain partnership with Singular directly benefits the DeFi Launch community. Here's an enhanced draft:

---

# Multichain Partnership Enhances Singular's Innovation and Growth
In the world of blockchain technology, partnerships are pivotal for driving growth and fostering innovation. Recently, Singular has joined forces with Multichain to bring exciting opportunities and solutions to their incubation program. This collaboration promises a robust foundation for Singular projects, enhancing their scalability, security, and interoperability.
### The Power of Multichain
Multichain is a leading blockchain platform that supports multiple digital assets and applications on a single network. This versatility offers significant benefits, including seamless cross-chain interoperability, enhanced scalability, and secure transactions across different blockchain networks. By integrating Multichain, Singular can offer its incubated projects a powerful ecosystem to grow and interact with other digital assets and networks.
### Benefits for Singular and Bikini Finance
#### 1. **Increased Interoperability**Singular's incubation program specializes in incubating projects that bridge the gap between blockchain technology and traditional financial systems. Multichain's interoperability feature enables Singular to easily integrate and exchange data and assets across different blockchain networks. This increased interoperability facilitates more seamless and efficient transactions, enriching the ecosystem for participating projects.
##### **Bikini Finance Overview**Bikini Finance, one of Singular's incubated projects, is a DeFi platform focused on enhancing liquidity and reducing trading friction. The project aims to leverage the Multichain ecosystem to develop and deploy stablecoin-generating swaps, decentralized exchanges, and other DeFi tools. These solutions aim to reduce trading costs and improve user experience, making DeFi more accessible to a wider audience.

### Creidt Assignment Analysis

### Worst Turn: #1

It completely misunderstands the relationship defined in the Target Document.

It claims Singular is the incubator (Target Doc says DeFi Launch is the incubator) and fails to mention any specific entities like Bikini Finance or tokens.

**ITPO Correct** | **Norm-ITPO Correct**

*Figure 16.* Part #1 of low-performance trajectory #2 in Example #1

#### 2. **Enhanced Scalability**Multichain's cross-chain infrastructure can significantly reduce the limitations of blockchain scalability. By ensuring smooth communication between different blockchain networks, holders of Bikini Finance can develop cutting-edge applications and decentralized finance (DeFi) tools that can handle a higher volume of transactions without compromising on security or speed.

##### **DeFi Launch Community**The partnership between Multichain and Bikini Finance directly benefits the DeFi Launch community, which is a significant target audience for Singular's incubation efforts. Through Multichain, Bikini Finance can tap into a larger user base and network of developers, thereby accelerating the launch and adoption of its projects.

#### 3. **Security and Compliance**Participating in the Multichain ecosystem ensures that Bikini Finance maintains high standards of security and compliance. The use of a unified platform like Multichain minimizes the risk of vulnerabilities and ensures that all projects adhere to strict security protocols, thereby building trust among users.

#### 4. **Developer Ecosystem Expansion**By integrating with Multichain, Bikini Finance can extend its reach to a broader developer community. This not only accelerates innovation but also attracts more projects to the Multichain ecosystem. Developers with access to Multichain have the freedom to choose the best blockchain solutions for their projects, potentially leading to faster development cycles and more diverse applications.

### Future Prospects
The partnership between Multichain and Singular is poised to revolutionize the blockchain ecosystem. By combining the strengths of these two platforms, Singular is not just incubating but also catalyzing innovation. The multichain framework enables Singular to foster a collaborative environment where ideas can intersect and converge, ultimately leading to groundbreaking applications and solutions.

### Summary
Incorporating Multichain into Singular's incubation program and partnerships, like Bikini Finance, brings numerous benefits. These include increased interoperability, enhanced scalability, improved security, and expanded developer ecosystems. For the DeFi Launch community, this partnership opens up new opportunities for faster and more secure blockchain solutions, potentially leading to widespread adoption and innovation.

---

This version should give a clearer picture of how the partnership directly benefits the DeFi Launch community and highlights Bikini Finance's upcoming projects within that ecosystem.

---

Great job expanding on Bikini Finance and the projects. Could you refine the section that explains the direct benefits for the DeFi Launch community a bit more? Specifically, I'd love to see clearer points on how this partnership leads to new farm and pool opportunities involving DLAUNCH, fDLAUNCH, and SING tokens. And maybe add some specifics on how these benefits translate into tangible improvements for community members?

---

Absolutely, let's refine the section on how the Multichain partnership with Singular and Bikini Finance directly benefits the DeFi Launch community, especially focusing on farm and pool opportunities and the tangible improvements for community members.

---

# Multichain Partnership Enhances Singular's Innovation and Growth
In the world of blockchain technology, partnerships are pivotal for driving growth and fostering innovation. Recently, Singular has joined forces with Multichain to bring exciting opportunities and solutions to their incubation program. This collaboration promises a robust foundation for Singular projects, enhancing their scalability, security, and interoperability.

### The Power of Multichain
Multichain is a leading blockchain platform that supports multiple digital assets and applications on a single network. This versatility offers significant benefits, including seamless cross-chain interoperability, enhanced scalability, and secure transactions across different blockchain networks. By integrating Multichain, Singular can offer its incubated projects a powerful ecosystem to grow and interact with other digital assets and networks.

### Benefits for Singular and Bikini Finance
#### 1. **Increased Interoperability**Singular's incubation program specializes in incubating projects that bridge the gap between blockchain technology and traditional financial systems. Multichain's interoperability feature enables Singular to easily integrate and exchange data and assets across different blockchain networks. This increased interoperability facilitates more seamless and efficient transactions, enriching the ecosystem for participating projects.

##### **Bikini Finance Overview**Bikini Finance, one of Singular's incubated projects, is a DeFi platform focused on enhancing liquidity and reducing trading friction. The project aims to leverage the Multichain ecosystem to develop and deploy stablecoin-generating swaps, decentralized exchanges, and other DeFi tools. These solutions aim to reduce trading costs and improve user experience, making DeFi more accessible to a wider audience.

#### 2. **Enhanced Security and Compliance**Participating in the Multichain ecosystem ensures that projects like Bikini Finance maintain high standards of security and compliance. The use of a unified platform like Multichain minimizes the risk of vulnerabilities and ensures that all projects adhere to strict security protocols, thereby building trust among users.

#### 3. **Direct Benefits for DeFi Launch Community**The partnership between Multichain and Singular introduces new farm and pool opportunities involving DLAUNCH, fDLAUNCH, and SING tokens, directly benefiting the DeFi Launch community:

##### **DLAUNCH and fDLAUNCH Tokens**DLAUNCH is a utility token for the DeFi Launch ecosystem, while fDLAUNCH is a staked version of the DLAUNCH token. These tokens provide community members with a stake in the platform's development and governance. By enabling the use of DLAUNCH and fDLAUNCH in various Multichain-based DeFi pools, stakeholders can earn rewards and participate in the community, gaining a direct financial stake in the platform's success.

##### **SING Token Integrations**Andersen SING, the SING token, can be used in various DeFi projects, including decentralized exchanges and yield farming mechanisms. Integrating SING tokens into the Multichain ecosystem allows for seamless cross-chain transactions, benefiting DeFi Launch community members who can leverage these tokens across different blockchain networks.

#### 4. **Developer Ecosystem Expansion**By integrating with Multichain, Bikini Finance and other Singularity-supported projects can extend their reach to a broader developer community. This not only accelerates innovation but also attracts more projects to the Multichain ecosystem. Developers with access to Multichain have the freedom to choose the best blockchain solutions for their projects, potentially leading to faster development cycles and more diverse applications.

### Future Prospects
The partnership between Multichain and Singular is poised to revolutionize the blockchain ecosystem. By combining the strengths of these two platforms, Singular is not just incubating but also catalyzing innovation. The multichain framework enables Singular to

---

**Creidt Assignment Analysis**

**Best Turn: #2**

It is the "cleanest" draft. It incorporates the requested Bikini Finance information without inventing fake names like "Andersen."

While it still shares the common error of misidentifying the incubator, it avoids the specific factual fabrications present in Responses 3 and 4.

---

*Figure 17.* Part #2 of low-performance trajectory #2 in Example #1

foster a collaborative environment where ideas can intersect and converge, ultimately leading to groundbreaking applications and solutions.
### Summary
Incorporating Multichain into Singular's incubation program and partnerships, like Bikini Finance, brings numerous benefits. These include increased interoperability, enhanced scalability, improved security, and expanded developer ecosystems. For the DeFi Launch community, this partnership opens up significant farm and pool opportunities involving DLAUNCH, fDLAUNCH, and SING tokens, providing tangible financial rewards and direct governance stakes. These improvements directly benefit community members by fostering a more active and inclusive ecosystem.

---
I hope this refines the content to provide a clearer picture of how the partnership directly benefits the DeFi Launch community and their community members.

Great work so far! Just to make sure, can you clarify how the new farm and pool opportunities with DLAUNCH, fDLAUNCH, and SING tokens specifically improve the daily life and activities of community members? I want to make sure the article clearly communicates these benefits and how they directly impact users in practical terms.

# Multichain Partnership Enhances Singular's Innovation and Growth
In the world of blockchain technology, partnerships are pivotal for driving growth and fostering innovation. Recently, Singular has joined forces with Multichain to bring exciting opportunities and solutions to their incubation program. This collaboration promises a robust foundation for Singular projects, enhancing their scalability, security, and interoperability.
### The Power of Multichain
Multichain is a leading blockchain platform that supports multiple digital assets and applications on a single network. This versatility offers significant benefits, including seamless cross-chain interoperability, enhanced scalability, and secure transactions across different blockchain networks. By integrating Multichain, Singular can offer its incubated projects a powerful ecosystem to grow and interact with other digital assets and networks.
### Benefits for Singular and Bikini Finance
#### 1. **Increased Interoperability**Singular's incubation program specializes in incubating projects that bridge the gap between blockchain technology and traditional financial systems. Multichain's interoperability feature enables Singular to easily integrate and exchange data and assets across different blockchain networks. This increased interoperability facilitates more seamless and efficient transactions, enriching the ecosystem for participating projects.
##### **Bikini Finance Overview**Bikini Finance, one of Singular's incubated projects, is a DeFi platform focused on enhancing liquidity and reducing trading friction. The project aims to leverage the Multichain ecosystem to develop and deploy stablecoin-generating swaps, decentralized exchanges, and other DeFi tools. These solutions aim to reduce trading costs and improve user experience, making DeFi more accessible to a wider audience.
#### 2. **Enhanced Security and Compliance**Participating in the Multichain ecosystem ensures that projects like Bikini Finance maintain high standards of security and compliance. The use of a unified platform like Multichain minimizes the risk of vulnerabilities and ensures that all projects adhere to strict security protocols, thereby building trust among users.
#### 3. **Direct Benefits for DeFi Launch Community**The partnership between Multichain and Singular introduces new farm and pool opportunities involving DLAUNCH, fDLAUNCH, and SING tokens, directly benefiting the DeFi Launch community. These benefits translate into tangible improvements for community members, enhancing their daily lives and activities:
##### **DLAUNCH and fDLAUNCH Tokens**DLAUNCH is a utility token for the DeFi Launch ecosystem, while fDLAUNCH is a staked version of the DLAUNCH token. When community members hold DLAUNCH and fDLAUNCH tokens, they can participate in various farm and pool opportunities:
- **Farm Opportunities**: DLAUNCH and fDLAUNCH tokens can be staked in DeFi farms, earning rewards for participating in liquidity provision and other staking activities. These rewards can be used to purchase more tokens, which in turn generate additional rewards, effectively forming a feedback loop that benefits token holders.- **Pool Participation**: DLAUNCH tokens can be used to generate yields in various AMM (Automated Market Maker) pools, such as Uniswap, thus increasing the community's liquidity and earning them rewards from portfolio growth.
##### **SING Token Integrations**Andersen SING, the SING token, can be used in various DeFi projects, including decentralized exchanges and yield farming mechanisms. Integrating SING tokens into the Multichain ecosystem allows for seamless cross-chain transactions, benefiting DeFi Launch community members who can leverage these tokens across different blockchain networks.
#### 4. **Developer Ecosystem Expansion**By integrating with Multichain, Bikini Finance and other Singularity-supported projects can extend their reach to a broader developer community. This not only accelerates innovation but also attracts more projects to the Multichain ecosystem. Developers with access to Multichain have the freedom to choose the best blockchain solutions for their projects, potentially leading to faster development cycles and more diverse applications.

**Creidt Assignment Analysis**

**Trajectory Score: 0.11**

**Low-Performance Trajectory #2 of Example #1**

*Figure 18.* Part #3 of low-performance trajectory #2 in Example #1

Can you help me write something about a Multichain partnership with Singular? I'm looking for something that highlights its benefits and how it fits into an incubation program.

Of course! I can help you craft a clear and engaging piece about a Multichain partnership with Singular. Let's break it down into a compelling paragraph that highlights the benefits and their integration into an incubation program.

---

In a significant move towards fostering innovation among blockchain platforms, Multichain has partnered with Singular to create a groundbreaking incubation program. By integrating Singular's unique solutions and expertise, this collaboration aims to accelerate the development of smart contracts, decentralized applications (dApps), and secure data exchanges across various blockchain ecosystems.
**Key Benefits:**
1. **Cross-Chain Compatibility**: Multichain's ability to support multiple blockchains enables Singular to seamlessly integrate its services across different platforms, breaking down barriers and offering greater flexibility in network operations.
2. **Enhanced Security**: Multichain's robust security mechanisms and Singular's advanced cryptographic techniques enhance the overall security landscape, safeguarding applications from potential vulnerabilities and ensuring data integrity.
3. **Faster Prototyping and Deployment**: This partnership facilitates quicker development cycles, allowing Singular's teams to prototype and deploy solutions more efficiently, thereby accelerating innovation and product adoption.
4. **Unified User Experience**: By harmonizing Singular's interfaces and functionalities with Multichain's network, the partnership provides a unified user experience, simplifying access and interaction for both developers and users.
In the context of an incubation program, this collaboration ensures that participants have access to state-of-the-art tools and expertise, fostering a robust ecosystem for startups and established companies alike. Multichain's expertise in network interoperability and Singular's cutting-edge solutions create an ideal environment for the incubation program to incubate and grow innovative blockchain based projects.

---

Feel free to use this as a starting point and add more details or specific examples as needed to make it more tailored to your needs.

Thanks for the draft! It looks good but could you tailor it more towards the benefits for the DeFi Launch community specifically? I want to emphasize how this partnership strengthens their ecosystem and supports their projects across different chains. Also, maybe mention Bikini Finance and the new farm and pool opportunities briefly?

Certainly! Here's a tailored version of the paragraph that emphasizes the benefits for the DeFi Launch community and highlights Bikini Finance and the new farm and pool opportunities:

---

In a significant move towards fostering innovation and fostering ecosystem collaboration among blockchain platforms, Multichain has partnered with Singular to create a groundbreaking incubation program designed specifically for the DeFi Launch community. This partnership aims to accelerate the development of scalable, secure, and interoperable DeFi solutions, ensuring a robust and unified ecosystem for both established and emerging projects.
**Key Benefits:**
1. **Cross-Chain Compatibility**: Multichain's ability to support multiple blockchains enables Singular to seamlessly integrate DeFi solutions across different platforms, breaking down barriers and offering greater flexibility in network operations. This is particularly advantageous for DeFi Launch projects, which often have cross-chain requirements to unlock global opportunities.
2. **Enhanced Security**: Multichain's robust security mechanisms and Singular's advanced cryptographic techniques enhance the overall security landscape, safeguarding DeFi applications from potential vulnerabilities and ensuring data integrity. This is critical for DeFi Launch projects, which are built on distributed networks and must prioritize security to attract and retain users.
3. **Faster Prototyping and Deployment**: This partnership facilitates quicker development cycles, allowing DeFi Launch teams to prototype and deploy solutions more efficiently. DeFi applications in different chains can now be integrated more seamlessly, reducing the time to market and enabling innovative DeFi protocols to reach their full potential.
4. **Unified User Experience**: By harmonizing Singular's interfaces and functionalities with Multichain's network, the partnership provides a unified user experience, simplifying access and interaction for both users and developers. This is particularly beneficial for projects aiming to be interoperable across multiple chains.
5. **Incorporating Piecemeal Platforms**: With Bikini Finance as a critical part of the ecosystem, this partnership provides new farm and pool opportunities. Ekto Finance, OneFinity, and other DeFi initiatives can now benefit from Multichain's cross-chain capabilities, expanding their reach and enhancing liquidity for users.
In the context of an incubation program, this collaboration ensures that DeFi Launch projects have access to state-of-the-art tools and expertise, fostering a robust ecosystem for innovation and growth. Multichain's expertise in network interoperability and Singular's advanced solutions create an ideal environment for DeFi projects to incubate and scale, making it easier for innovative DeFi protocols to unlock global opportunities.

---

This version emphasizes how the partnership strengthens the DeFi Launch ecosystem and highlights the new opportunities for projects.

Thanks for updating the draft! Could you make sure Bikini Finance and the new farm and pool opportunities are even more prominent? Also, could you confirm if these opportunities involve $DLAUNCH and $fDLAUNCH at BSC and Fantom, respectively, along with $SING?

Absolutely! Let's make sure that Bikini Finance and the new farm and pool opportunities are prominently featured, and confirm the specific details you mentioned. Here's the refined version of the paragraph:

---

In a significant move towards fostering innovation and fostering ecosystem collaboration among blockchain platforms, Multichain has partnered with Singular to create a groundbreaking incubation program designed specifically for the DeFi Launch community. This partnership aims to accelerate the development of scalable, secure, and interoperable DeFi solutions, ensuring a robust and unified ecosystem for both established and emerging projects.

**Creidt Assignment Analysis**

**Worst Turn: #1**

It completely misses the context. It hallucinates a corporate partnership between the bridge protocol "Multichain" and "Singular," rather than writing from the perspective of DeFi Launch (the incubator). It is purely generic fluff with zero specific details (no mention of Bikini Finance, specific tokens, or specific chains) found in the target document.

**ITPO Correct**    **Norm-ITPO Correct**

*Figure 19.* Part #1 of high-performance trajectory #1 in Example #1

**Key Benefits:**
1. **Cross-Chain Compatibility**: Multichain's ability to support multiple blockchains enables Singular to seamlessly integrate DeFi solutions across different platforms, breaking down barriers and offering greater flexibility in network operations. This is particularly advantageous for DeFi Launch projects, which often have cross-chain requirements to unlock global opportunities.
2. **Enhanced Security**: Multichain's robust security mechanisms and Singular's advanced cryptographic techniques enhance the overall security landscape, safeguarding DeFi applications from potential vulnerabilities and ensuring data integrity. This is critical for DeFi Launch projects, which are built on distributed networks and must prioritize security to attract and retain users.    3. **Faster Prototyping and Deployment**: This partnership facilitates quicker development cycles, allowing DeFi Launch teams to prototype and deploy solutions more efficiently. DeFi applications in different chains can now be integrated more seamlessly, reducing the time to market and enabling innovative DeFi protocols to reach their full potential.
4. **Unified User Experience**: By harmonizing Singular's interfaces and functionalities with Multichain's network, the partnership provides a unified user experience, simplifying access and interaction for both users and developers. This is particularly beneficial for projects aiming to be interoperable across multiple chains.    5. **Incorporating Piecemeal Platforms**: With Bikini Finance as a critical part of the ecosystem, this partnership provides new farm and pool opportunities. Specifically, Bikini Finance's pool and farm mechanisms are now enhanced with Multichain's cross-chain capabilities, allowing for more seamless liquidity and investment across various blockchains. For example, $DLAUNCH will now be available as a farm token on BSC, while $fDLAUNCH will be available in pools on Fantom, alongside $SING.
In the context of an incubation program, this collaboration ensures that DeFi Launch projects have access to state-of-the-art tools and expertise, fostering a robust ecosystem for innovation and growth. Multichain's expertise in network interoperability and Singular's advanced solutions create an ideal environment for DeFi projects to incubate and scale, making it easier for innovative DeFi protocols to unlock global opportunities.
---
This version highlights Bikini Finance and the specific new farm and pool opportunities on BSC and Fantom, ensuring clarity and detail.

Thanks for the updates! It sounds better, but can you make sure the part about the new farm and pool opportunities involving $DLAUNCH, $fDLAUNCH, and $SING is as clear and concise as possible? I want to make it easy for readers to understand the benefits and opportunities they'll get from these partnerships. Also, can you briefly mention any additional incentives or perks for participating in these farms and pools?

---
In a significant move towards fostering innovation and fostering ecosystem collaboration among blockchain platforms, Multichain has partnered with Singular to create a groundbreaking incubation program designed specifically for the DeFi Launch community. This partnership aims to accelerate the development of scalable, secure, and interoperable DeFi solutions, ensuring a robust and unified ecosystem for both established and emerging projects.
**Key Benefits:**
1. **Cross-Chain Compatibility**: Multichain's ability to support multiple blockchains enables Singular to seamlessly integrate DeFi solutions across different platforms, breaking down barriers and offering greater flexibility in network operations. This is particularly advantageous for DeFi Launch projects, which often have cross-chain requirements to unlock global opportunities.
2. **Enhanced Security**: Multichain's robust security mechanisms and Singular's advanced cryptographic techniques enhance the overall security landscape, safeguarding DeFi applications from potential vulnerabilities and ensuring data integrity. This is critical for DeFi Launch projects, which are built on distributed networks and must prioritize security to attract and retain users.
3. **Faster Prototyping and Deployment**: This partnership facilitates quicker development cycles, allowing DeFi Launch teams to prototype and deploy solutions more efficiently. DeFi applications in different chains can now be integrated more seamlessly, reducing the time to market and enabling innovative DeFi protocols to reach their full potential.
4. **Unified User Experience**: By harmonizing Singular's interfaces and functionalities with Multichain's network, the partnership provides a unified user experience, simplifying access and interaction for both users and developers. This is particularly beneficial for projects aiming to be interoperable across multiple chains.
5. **Incorporating Piecemeal Platforms**:    - **Bikini Finance**: With Bikini Finance as a critical part of the ecosystem, this partnership enhances its pool and farm mechanisms with Multichain's cross-chain capabilities. For example, $DLAUNCH will now be available as a farm token on BSC, while $fDLAUNCH will be available in pools on Fantom, alongside $SING.
This version clearly outlines Bikini Finance's additional pools and farms, including specific tokens ($DLAUNCH and $fDLAUNCH), and explicitly mentions the additional incentives and perks associated with participating in these opportunities.

**Creidt Assignment Analysis**

**Best Turn: #3**

It correctly integrates Bikini Finance.
It correctly maps the tokens to their chains: $DLAUNCH on BSC and $fDLAUNCH on Fantom, along with $SING.

Unlike Response 2, it avoids hallucinating unrelated projects (like Ekto Finance).

**ITPO Correct** | **Norm-ITPO Correct**

**Trajectory Score: 0.29**

**High-Performance Trajectory #1 of Example #1**

*Figure 20.* Part #2 of high-performance trajectory #1 in Example #1

Can you help me write something about a Multichain partnership with Singular? I'm looking for something that highlights its benefits and how it fits into an incubation program.

Of course! Let's craft a compelling piece that highlights the benefits of a partnership between Multichain and Singular, and how it fits into an incubation program.

---
### Partnerships Spotlight: Multichain and Singular Prove a Powerful Duo for Innovation
In today's rapidly evolving blockchain landscape, staying ahead of the game requires groundbreaking collaborations. One such collaboration that has caught the attention of the blockchain community is the strategic partnership between **Multichain** and **Singular**. Their partnership is setting new benchmarks in interoperability and innovation by unlocking the full potential of blockchain technology for a wide range of industries.
#### What Are They About?
**Multichain** is leading the charge in creating an advanced blockchain framework that enables developers to build and scale blockchain applications. Their proprietary platform allows developers to create secure, interoperable, and scalable blockchain networks without the need to optimize their blockchains. By providing a high-speed, rock-solid, and fully-chain-as-a-service environment, **Multichain** removes the hardware or software constraints from the development process, making it more accessible and efficient.
**Singular** is an incubation program focused on fostering the development of next-generation digital identity solutions. Their innovative approach to digital identity ensures secure, scalable, and interoperable solutions that enhance trust and facilitate seamless access in various digital ecosystems.
#### The Benefits of Their Collaboration
By partnering with **Multichain**, **Singular** gains invaluable technical capabilities and tools essential for their incubation program. Here are some specific benefits:
1. **Interoperability**: Multichain's advanced platform allows seamless integration of Singular's digital identity solutions across different blockchain environments. This interoperability ensures that Singular's clients can enjoy secure and flexible digital identities regardless of the network they are on.
2. **Accelerated Development**: With **Multichain**'s support, Singular's incubation process is accelerated because the core foundational infrastructure for developing blockchain solutions is already in place. Developers don't need to worry about building the basic layers of a blockchain network, allowing them to focus on innovation and application development.
3. **Security and Reliability**: Multichain's commitment to security and reliability ensures that the blockchain solutions developed within the **Singular** incubation program are robust and resilient. This security can be a strong selling point for clients and investors, fostering trust and confidence in the innovative projects.
4. **Future-Proof Solutions**: Multichain's ability to scale and adapt to future blockchain developments means that the incubated solutions developed in this partnership will be well-equipped to face future technological advancements. This forward-thinking approach helps Singular and its clients stay ahead in an ever-evolving technological landscape.
#### How This Partnership Fits into an Incubation Program
The partnership between **Multichain** and **Singular** is not just a one-off deal but rather a strategic alliance designed to fuel innovation and growth. Here's how it can shape and support an incubation program:
1. **Accelerated Product Development**: The partnership streamlines the development cycle by providing the necessary technological platform and support. Incubation providers like Singular can focus on refining the user experience and scaling technical solutions more effectively.
2. **Enhanced Portfolio of Solutions**: With Multichain's backing, Singular's incubation portfolio becomes more robust and diverse, catering to specific market requirements across different industries. This can attract a wider range of clients and foster a strong market presence.
3. **Improved Incentives and Support**: Access to Multichain's state-of-the-art infrastructure and support services can provide significant benefits to the incubated projects, including reduced costs, enhanced scalability, and improved security. This, in turn, can benefit the incubation program by attracting more high-profile projects and expanding its offerings.
4. **Brand Credibility**: A strong partnership like this not only adds credibility to the incubation program but also builds stronger alliances in the blockchain ecosystem. This can lead to increased visibility, securing more funding, and partnerships with other key players in the industry.
In summary, the partnership between **Multichain** and **Singular** represents a powerful synergy that benefits both organizations. By leveraging Multichain's robust offerings and Singular's innovative incubation model, they can drive the development of transformative blockchain solutions and foster a thriving ecosystem of secure, scalable, and interoperable digital identity and application development.

---
Feel free to tweak the content to better fit the specific details of your project or use it as a base to create your own piece.

It looks like there's been a bit of a mix-up. Could you please rework the article to specifically highlight the partnership between Multichain and Singular regarding DeFi projects, mentioning Bikini Finance and the DeFi Launch community? Also, could you ensure that the article stays under 500 words?

Certainly! Here's a focused piece highlighting the partnership between Multichain and Singular, specifically with DeFi projects like Bikini Finance, and incorporating the DeFi Launch community.

---
### Partnerships Spotlight: Multichain, Singular, Bikini Finance, and DeFi Launch Collab
**Multichain** and **Singular** have formed a strategic partnership to support the development of DeFi projects, particularly focusing on **Bikini Finance** and the **Defi Launch** community. This collaboration aims to bring groundbreaking DeFi solutions to life and enhance interoperability within the DeFi ecosystem.
#### What Is Bikini Finance?
**Bikini Finance** is a decentralized finance (DeFi) platform that simplifies access to decentralized lending, borrowing, and trading by leveraging the Multichain platform. This partnership will enable **Bikini Finance** to launch its services more seamlessly and securely across different blockchain networks, expanding its user base and enhancing user experience.

**Creidt Assignment Analysis**

**Worst Turn: #1**

It suffers from a critical hallucination. The Target Document defines "Singular" as a yield farming protocol.

Response 1 completely invents a different identity for it, describing it as an "incubation program focused on digital identity solutions."

It also fails to mention key entities like Bikini Finance.

**ITPO Correct** **Norm-ITPO Correct**

*Figure 21.* Part #1 of high-performance trajectory #2 in Example #1

#### Enhanced Interoperability with Multichain
**Multichain** provides developers with a high-speed, secure, and scalable blockchain framework, making it easier for applications to connect and interact across different network environments. By using Multichain, **Bikini Finance** can integrate with a wide variety of DeFi protocols, ensuring seamless user experiences and unblocking potential use cases.
#### Fostering Innovation through Singular
**Singular**'s incubation program will actively support the development of innovative DeFi solutions through **Bikini Finance**. This collaboration will accelerate the process of building and launching new DeFi projects, focusing on security, scalability, and user-friendliness.
#### The Crucial Role of Defi Launch
**Defi Launch**, a community of DeFi developers and enthusiasts, plays a vital role in this partnership. The community provides a platform for projects like **Bikini Finance** to showcase their capabilities, receive feedback and support, and participate in a global network of distributed developers. By leveraging this community, Multichain and Singular can attract more startups and reputable DeFi projects to bootstrap their development efforts.
#### Benefits for Developers and Investors
- **Accelerated Development**: Multichain's advanced platform simplifies the development process, allowing developers to focus on core application functionality rather than foundational infrastructure.- **Security and Reliability**: The firm's focus on robust security measures ensures that **Bikini Finance** and other DeFi projects are built on stable, reliable foundations.- **Increased Value**: Enhanced interoperability and scalability drive competitive advantage, attracting more investors and users to the platform.
In summary, the partnership between **Multichain** and **Singular**, supported by **Bikini Finance** and the **Defi Launch** community, represents a transformative alliance in the DeFi sphere. By leveraging Multichain's powerful ecosystem and Singular's supportive incubation program, projects like **Bikini Finance** can flourish, leading to a more interconnected and accessible DeFi landscape.
---
This version of the article maintains a focus on the partnership and its benefits while staying under 500 words.

Great start, but I feel it misses out on the specifics about Singular being a yield farming protocol across multiple chains and the strategic partnership it brings to the table for DeFi Launch. Could you adjust the article to focus more on Singular's cross-chain capabilities and the opportunities for new farms and pools, especially those involving Bikini Finance and the DeFi Launch community?

Certainly! Let's expand on the specifics of Singular's cross-chain capabilities and its strategic partnership with DeFi Launch, highlighting how Bikini Finance fits into this ecosystem. Here's a refined version:
---
### Partnerships Spotlight: Multichain, Singular, Bikini Finance, and DeFi Launch Collab
**Multichain** and **Singular** have formed a strategic partnership to enhance the development of DeFi projects, especially those leveraging Bikini Finance and the DeFi Launch community. By integrating Singular's cross-chain capabilities, this collaboration aims to drive innovation and interoperability within the DeFi ecosystem.
#### The Power of **Singular**
**Singular** is a yield farming protocol that operates across multiple blockchain networks, ensuring that users can generate returns on their tokens regardless of the underlying blockchain. This cross-chain functionality allows for more stable and diversified returns, making it an attractive choice for users and developers alike.
One prominent example of this cross-chain capability is **Bikini Finance**, a DeFi platform that offers DeFi Launch members exclusive access to innovative DeFi projects. By utilizing MultiChain, **Bikini Finance** can integrate with numerous blockchain networks, enabling developers to build DeFi solutions that seamlessly interact across these networks.
#### Enhanced Interoperability with **Bikini Finance**
Through the partnership with **Singular**, **Bikini Finance** can leverage **Multichain** to enhance its user experience and integrate its DeFi protocols across a broader range of blockchain networks. This interoperability ensures that users can access a variety of DeFi features, including decentralized lending, borrowing, and token trading, without constraints.
#### Accelerating Innovation with **DeFi Launch**
**DeFi Launch** is a community of distributed DeFi developers and enthusiasts that provides a platform for projects to showcase their capabilities, receive feedback, and participate in a global network of innovators. By partnering with **Singular** and **Bikini Finance**, this community can benefit from the enhanced ecosystem driven by the combined efforts of these partners.
- **Cross-Chain Farms and Pools**: By integrating **Multichain** and **Singular**, **Bikini Finance** can create cross-chain farming and liquidity pools. For example, users can stake tokens on **Bikini Finance** and earn rewards on different chains, benefiting from the stability and liquidity provided by **Singular**.
- **Unique Project Showcase**: **DeFi Launch** members can now showcase their projects within a secure, interoperable environment facilitated by **Singular**. This not only provides a platform for project pitch competitions but also facilitates collaboration, fostering a thriving ecosystem of DeFi solutions.
#### The Strategic Alliance
- **Faster Development**: With **Singular**'s support, projects like **Bikini Finance** can focus on innovation rather than foundational infrastructure, allowing for more rapid development cycles.- **Stable Rewards**: Cross-chain farming means that users can earn stable returns on their tokens, ensuring consistent income across multiple chains, enhancing liquidity and reducing risks.- **Enhanced Security**: By leveraging **Multichain**'s strong security framework, all projects within the ecosystem can benefit from robust protections, ensuring that DeFi solutions are built on secure ground.
In summary, the partnership between **Multichain**, **Singular**, **Bikini Finance**, and the DeFi Launch community represents a transformative alliance in the DeFi sphere. This collaboration leverages **Singular**'s cross-chain capabilities and **Multichain**'s robust infrastructure, driving innovation and providing a secure, interconnected ecosystem for next-generation DeFi projects.
---

Creidt Assignment Analysis

*Figure 22.* Part #2 of high-performance trajectory #2 in Example #1

Great work so far, but I think we need to dial up the focus on the new farms and pools. Specifically, I'd love to see more detail about how these new opportunities will involve Bikini Finance and the DeFi Launch community. Additionally, could you mention how $DLAUNCH and $SING will play into these new initiatives? I think emphasizing these points will make the article more engaging and informative.

Absolutely, let's focus more on the new farms and pools that will involve Bikini Finance and the DeFi Launch community, and specifically highlight the roles of $DLAUNCH and $SING tokens. Here's a revised version:

---

### Partnerships Spotlight: Multichain, Singular, Bikini Finance, and DeFi Launch Collab

**Multichain**, **Singular**, **Bikini Finance**, and the **DeFi Launch** community are forging a powerful alliance to drive innovation and interconnective DeFi projects. By integrating **Singular**'s cross-chain capabilities, this partnership aims to create new DeFi opportunities that are accessible, secure, and interoperable across multiple blockchain networks.

#### The Power of **Singular**

**Singular** is a yield farming protocol that operates across multiple blockchain networks, providing users with stable and diversified returns on their tokens. This cross-chain functionality ensures that users can earn rewards on a variety of tokens without being limited to a single blockchain. One notable example of this is **Bikini Finance**, which is a DeFi platform that offers exclusive access to DeFi Launch members.

#### New Farms and Pools Involving Bikini Finance and DeFi Launch

Through **Singular**'s enhanced ecosystem, **Bikini Finance** can create unique farming and liquidity pools that integrate seamlessly across multiple blockchain networks. These new opportunities will involve the **DeFi Launch** community and their members, offering a variety of cross-chain farm and liquidity pool configurations.
- **$DLAUNCH and $SING Staking**: New cross-chain farming and liquidity pools leveraging **Singular** will reward users with tokens like **$DLAUNCH** and **$SING**, which can be staked to earn additional rewards. Users can choose to stake their tokens on a pool within **Bikini Finance** and earn rewards on multiple chains simultaneously.
- **DeFi Launch Members**: DeFi Launch members, with their exclusive access, can offer exclusive pools or incentives for their community members, driving participation and fostering a collaborative ecosystem. These members can tap into the **Bikini Finance** platform to create custom pools or earn rewards for staking tokens within various DeFi projects.

#### Enhanced Security and Stability

By leveraging **Multichain**'s robust security framework, these new initiatives ensure that DeFi solutions are built on secure ground. **Multichain**'s platform provides developers with a high-speed, scalable, and interoperable environment, ensuring that DeFi projects can exist and interact on multiple networks without constraints.

#### Faster Development and Collaborative Ecosystem

- **Faster Development**: With **Singular**'s support, projects like **Bikini Finance** can focus on innovation rather than foundational infrastructure, allowing for more rapid development cycles.- **Stable Rewards**: Users can earn stable rewards on a variety of token pools across different chains, enhancing liquidity and reducing risks.- **Community Driven**: The DeFi Launch community can participate in creating and managing these new pools, fostering a collaborative ecosystem where innovation drives growth.

#### Key Tokens Involved

- **$DLAUNCH (DeFi Launch Token)**: This token is the governance token for the DeFi Launch community. Members can participate in community decisions, voting on pool configurations, and other governance matters.- **$SING (Singular Token)**: This token represents ownership in **Singular** and is used to staking rewards on cross-chain pools.

#### Conclusion

The partnership between **Multichain**, **Singular**, **Bikini Finance**, and the **DeFi Launch** community represents a transformative alliance in the DeFi sphere. By leveraging **Singular**'s cross-chain capabilities and **Multichain**'s robust ecosystem, **Bikini Finance** can create new and exciting DeFi opportunities. With the support of the DeFi Launch community, these new initiatives will drive innovation, attract more users and investors, and foster a thriving DeFi ecosystem.

---

This version of the article focuses more on the specifics of new farms and pools, emphasizing the roles of $DLAUNCH and $SING, and how they fit into the overall ecosystem. It also highlights the value propositions and key tokens involved.

**Creidt Assignment Analysis**

**Best Turn: #4**

Unlike the first two responses, it correctly identifies Singular as a yield farming protocol. Completeness: It is the only response that captures the specific token details mentioned in the Target Document's conclusion. It explicitly names $DLAUNCH, $fDLAUNCH, and $SING within the context of the new farms and pools.
It correctly links Multichain (as a capability/platform), Singular, Bikini Finance, and the DeFi Launch community together in the way the target document intends.

| ITPO Correct | Norm-ITPO Correct |

**Trajectory Score: 0.19**

**High-Performance Trajectory #2 of Example #2**

*Figure 23.* Part #3 of high-performance trajectory #2 in Example #1

Above are the first $4$ examples with same initial prompts for turn-wise reward allocation interpretability analysis. For the rest $28$ examples, please refer to supplementary materials.

