# OpenReview forum: "Implicit Turn-Wise Policy Optimization for Proactive User-LLM Interaction"
_ICML.cc/2026/Conference — ICML 2026 regular_

### Official Review · Reviewer_yLeh · 2026-03-12

**Soundness:** 3
**Presentation:** 3
**Significance:** 3
**Originality:** 3
**Overall Recommendation:** 5
**Confidence:** 3

**Summary:**

This paper addresses the problem of reward sparsity in multi-turn human-LLM interactions by introducing Implicit Turn-wise Policy Optimization (ITPO). The method aggregates token-level implicit process rewards into turn-level signals, utilizing a normalization mechanism to align the scale of these intermediate rewards with the final outcome reward. The authors evaluate ITPO across three diverse collaborative tasks (Math Tutoring, Document Writing, and Medical Recommendation), demonstrating consistent improvements over standard trajectory-level and token-level reward assignment baselines when paired with standard advantage estimators like PPO, GRPO, and RLOO.

**Compliance With Llm Reviewing Policy:**

Affirmed.

**Final Justification:**

After considering the rebuttal and discussion, I am satisfied with the authors’ responses. Overall, I believe the paper is suitable for acceptance at ICML. I maintain my score at 5.

**Key Questions For Authors:**

1. Could the significant reduction in token usage for ITPO (as seen in Table 1) be the primary driver of the improved overall score due to the length penalty? How does the method perform if the length penalty is entirely removed during training?
2. In Figure 7, the mapping slope between the implicit reward and the outcome reward tends to increase over time. Do you have a hypothesis or theoretical intuition for why the implicit PRM naturally drifts in this specific direction during online optimization?
3. Why was BLEU chosen as the primary metric for the Document Writing task, given its known limitations in evaluating long-form LLM generation? Did you observe different trends when looking at semantic similarity or LLM-based evaluation?
4. How sensitive is the ITPO method to the specific choice of the reference model \pi_ref? Does performance degrade if the reference model is smaller or less capable than the policy model?

**Limitations:**

Add a brief discussion on the limitations of training against an LLM-based user simulator, particularly concerning the sim-to-real gap when deploying such proactive agents to real human users.

**Strengths And Weaknesses:**

Strengths: The motivation to shift from high-variance token-level implicit rewards to turn-level rewards is intuitive and well-supported by the problem domain of multi-turn dialogue. The proposed normalization mechanism elegantly addresses the scale instability between implicit rewards and the true outcome reward. The combination of the proposed reward formulation with multiple standard RL algorithms (PPO, GRPO, RLOO) shows that the method is agnostic to the underlying policy optimizer. The evaluation spans three distinct and challenging multi-turn interaction scenarios.

Weakness:
1. The reliance on BLEU for the Document Writing task severely limits the reliability of those specific results. Generative tasks in multi-turn interactions require semantic evaluation, and BLEU is widely considered inadequate for assessing the quality of LLM-generated documents.
2. The human evaluation of interpretability in Section 4.3.2 is based on a very small subset (32 trajectories). While the results show good alignment with human judgment, this sample size is quite small for drawing robust statistical conclusions about the method's interpretability.
3. The evaluation environment assumes the LLM user simulator accurately reflects the latent goals of real users. The variance of real human responses is arguably much higher than what an LLM simulator produces. The paper should discuss the robustness of the learned implicit rewards under highly out-of-distribution human interactions.

---

> ### Author Rebuttal · Authors · 2026-03-31
>
> We thank reviewer yLeh for raising the insightful questions. Below we try to address their questions.
>
> Q1: Performance without length penalty
> ---
> (1) Given the limited resources, we run experiments on math tutoring (which result in the most significant token drop), with only accuracy as reward. The training dynamic is in [(Sec. 3 in link)](https://anonymous.4open.science/r/supplementary_figures-3F3B), and results are below:
>
> | Method | Accuracy |
> |------------------|---------:|
> | Trajectory-Level | 28.67 |
> | ITPO | 35.33 |
> | Norm-ITPO | 38.00 |
>
> ITPO still improves performance, indicating that its gain is not simply due to shorter outputs.
>
> (2) The token-number regularization is meaningful for real-world applications. For commercial LLMs, reducing token usage while improving quality is an important goal. In this sense, including a length penalty is aligned with the intended application scenario rather than being an artificial design choice.
>
> We would include the corresponding ablation results.
>
> Q2: The mapping slope in Figure 7
> ---
> We thank the reviewer for this insightful question. The mapping slopes tend to first increase and then plateau throughout training. One possible intuition is: during early optimization, the RM becomes more confident in distinguishing preferred trajectories from less preferred ones. Its effective margin between good and bad trajectories increases, which results in a steeper mapping to the outcome reward.
>
> Q3 & W1: BLEU score
> ---
> We thank the reviewer for pointing this out.
>
> (1) We use BLEU for Doc Writing mainly to stay consistent with CollabLLM[1], from which this task is adopted. CollabLLM evaluates the doc writing setting with BLEU, measuring similarity between the edited document and the target.
>
> (2) We agree that BLEU has well-known limitations for evaluating long-form LLM generation. To address this concern, we additionally train the doc writing task with LLM-as-a-judge protocol, and we observe a similar trend. The training dynamics and LLM judge prompts are in [(Sec. 4 in link)](https://anonymous.4open.science/r/supplementary_figures-3F3B), results are below.
>
> | Method | Judge Reward | #Token(k) | Score |
> |------------------|-----------------:|----------:|------:|
> | Trajectory-Level | 63.23 | 0.75 | 59.46 |
> | ITPO | 67.34 | 0.94 | 62.61 |
> | Norm-ITPO | 69.76 | 0.96 | 64.89 |
>
> [1] CollabLLM: From Passive Responders to Active Collaborators.Wu, et al. ICML 2025.
>
> Q4: Choice of Reference Model
> ---
> We thank the reviewer for this insightful question. In our setup, $\pi_{ref}$ is initialized as the same as the policy. We did not vary the reference model as an independent design axis.
>
> Prior work suggests that implicit PRM is not overly sensitive to $\pi_{ref}$[1], the reference model can even be changed or omitted without hurting accuracy. Since ITPO builds on similar formulation, this provides indirect evidence that ITPO may also be relatively insensitive to the reference choice, though this remains an extrapolation rather than a direct result from our paper.
>
> [1] Free Process Rewards Without Process Labels. Yuan, et al. ICML 2025.
>
> W2: Human evaluation
> ---
> We agree that human evaluation is based on a relatively small sample and should be viewed as a limitation.
>
> The current scale includes 32 trajectories, with 64 human decisions in total, since each contributes a best and worst label. ITPO and Norm-ITPO achieve higher agreement (47/48 out of 64) with human judgments than random choice (16 of 64), providing preliminary evidence that the turn-level implicit rewards are aligned with human preferences.
>
> Meanwhile, each trajectory is relatively long (see the appendix examples), making annotation costly and discussion-intensive. We would revise the manuscript to present this study as a pilot-scale human evaluation, and we view large-scale benchmarking for multi-turn RM interpretability as a future direction.
>
> W3: Gap Between Simulator and Human
> ---
> We thank the reviewer for this important question. We try to address this from two perspectives.
>
> (1) We test robustness under a different simulator/judge. Please refer to response to **Reviewer URbJ [Q1]** for details. ITPO and Norm-ITPO still outperform the baseline, suggesting that the method is not overly dependent on a single simulator pattern.
> At this time, we agree that cross-simulator robustness is not entirely equivalent to robustness under real human interaction.
>
> (2) We acknowledge that closing the gap between simulated and real users is an important open problem. We would revise the manuscript to make this limitation explicit and clarify that the current evidence supports robustness across simulators, but does not yet establish robustness under highly OOD real users.
>
> Limitations:
> ---
> We thank the reviewer for this helpful suggestion. We will include a corresponding discussion in the limitations section of the final version. Please refer to our response to  **Reviewer URbj [Limitation]** for the full revised limitation.

---

> > ### Author Rebuttal · Reviewer_yLeh · 2026-04-07
> >
> > Thanks to the authors for the clarifications.

---

### Official Review · Reviewer_C4CU · 2026-03-12

**Soundness:** 3
**Presentation:** 3
**Significance:** 3
**Originality:** 2
**Overall Recommendation:** 4
**Confidence:** 4

**Summary:**

This paper proposes an implicit reward method for multi-turn tasks, which effectively addresses the issue of sparse rewards and can be applied as a plug-and-play module across various policy optimization algorithms.

**Compliance With Llm Reviewing Policy:**

Affirmed.

**Final Justification:**

The author's reply resolved part of my questions. Overall, I maintain my original rating.

**Key Questions For Authors:**

No

**Limitations:**

yes

**Strengths And Weaknesses:**

Strengths:
1. The implicit reward calculation method proposed in the paper is concise, effective, and easy to reproduce.
2. The normalization method proposed in the paper effectively addresses the issue of magnitude differences.
3. The method can be integrated with popular policy optimization algorithms, demonstrating good generality.

Weaknesses:
1. Norm-ITPO lacks a solid theoretical foundation. As a concept similar to the implicit reward in DPO, we would expect it to be accompanied by a mathematical derivation based on RLHF objective function. Although the paper offers a Bayesian-based explanation, it feels somewhat forced and is not closely related to reinforcement learning.

2. For algorithms like GRPO, each group of samples should originate from the same state ($ h_k $). However, in the paper's application, it seems that samples with the same timestep are simply clustered into the same group (according to the formula in Section 3.4), which is inconsistent with GRPO. If the authors' actual approach differs, it is recommended to provide a clearer description to avoid misunderstandings.

3. The experimental results show weaker performance on some metrics, such as the Token(k) for Doc Writing.

---

> ### Author Rebuttal · Authors · 2026-03-31
>
> We thank **reviewer C4CU** for their constructive feedback and insightful comments. Below we try to explain the weaknesses:
>
> W1: Theoretical foundation of Norm-ITPO.
> ---
> We thank the reviewer for this thoughtful and fair comment. We agree that Norm-ITPO currently does not have a theoretical foundation derived from RL perspective comparable to that of ITPO. While ITPO can be motivated through the implicit PRM formulation and its theoretical pipeline, Norm-ITPO was introduced primarily from empirical observations: during training, we found that implicit reward might encounter stability issue(see Figure. 7), which led us to propose Norm-ITPO as a practical heuristic.
>
> Potential thoughts on deriving Norm-ITPO from the RL perspective might analyze the structural properties induced by the normalization. For example, whether the normalized turn-wise rewards preserve key trends of the original implicit rewards, such as monotonicity, and relative ranking consistency.
>
> In the final version, we will revise this part to more clearly separate what is theoretically derived from what is empirically motivated, and present Norm-ITPO in a more accurate way as a heuristic variant rather than a fully derived objective.
>
> W2: Clarification on GRPO Notations
> ---
> We thank the reviewer for pointing this out. The confusion comes from our notation in Sec. 3.4.
>
> In ITPO, the advantage calculation does not group by timestep across different initial prompts. Instead, it uses a GRPO-style normalization: for all rollout samples generated from the same initial prompt, we collect the implicit process rewards from all turns across the rollouts, and use their mean and standard deviation to normalize the turn-level reward.
>
> Given one initial prompt, the advantage of the $i$-th sample at the $k$-th turn is
>
> $$
> A_{i,k}^{\mathrm{GRPO}} = \frac{R_{i,k}^{\phi} - \mu}{\sigma + \epsilon}.
> $$
>
> The normalization statistics are computed as
>
> $$
> \mu = \frac{1}{|\mathcal{G}|} \sum_{(i,k)\in\mathcal{G}} R_{i,k}^{\phi},
> $$
>
> $$
> \sigma = \text{Std of } R_{i,k}^{\phi} \text{ over } (i,k)\in\mathcal{G}.
> $$
>
> Here, $\mathcal{G}$ denotes the set of all turn-level samples from all rollouts associated with the same initial prompt. Therefore, $\mu$ and $\sigma$ are the mean and standard deviation of the implicit process rewards within this prompt-level group.
>
> We would revise the manuscript to make this point clearer and avoid misunderstanding.
>
> W3: The experimental results show weaker performance on some metrics, such as the Token(k) for Doc Writing.
> ---
> We thank the reviewer for pointing this out. We agree that ITPO does not outperform the baselines on every individual metric for every task, such as Token(k) in Doc Writing.
>
> For Doc writing, one possible explanation is that a higher-quality response under the overall scoring function may require more detailed or comprehensive content, which can naturally increase output length, leading to the trade-off between doc quality and doc length.  And ITPO improves the overall objective, but it does not guarantee uniform gains on every evaluation dimension.
> We will clarify this trade-off in the revised manuscript.
>
> More broadly, **reviewer C4CU** pointed out an interesting and urgent direction: balancing multiple potentially competing objectives is an important direction for future work in multi-objective RL.

---

> > ### Author Rebuttal · Reviewer_C4CU · 2026-04-01
> >
> > Thank you for your reply. My questions have been partially answered. Regarding the response to W1, since the author did not provide a formal proof and derivation process, I am unable to determine whether they have addressed my concern. I will maintain my current rating.

---

### Official Review · Reviewer_URbJ · 2026-03-12

**Soundness:** 3
**Presentation:** 3
**Significance:** 3
**Originality:** 3
**Overall Recommendation:** 5
**Confidence:** 3

**Summary:**

The paper studies reinforcement learning for multi-turn user–LLM interactions where supervision often comes only from a final outcome reward, making credit assignment across turns difficult. It proposes Implicit Turn-wise Policy Optimization (ITPO), which learns an implicit process reward signal from outcome feedback and aggregates it at the turn level so each dialogue turn becomes the main unit for learning. A variant, Norm-ITPO, aims to stabilize training by distributing outcome credit across turns in a normalized way. The approach is designed to be compatible with PPO-style training and is evaluated on math tutoring, document writing, and medical recommendation settings using user simulation and automatic judging, showing consistent improvements over several baselines.

**Compliance With Llm Reviewing Policy:**

Affirmed.

**Final Justification:**

Increased scores.

**Key Questions For Authors:**

- How robust are the results to different user simulators and different outcome judges?
- Do you have any evidence for transfer to real users, or to evaluation based on human data / human ratings?

**Limitations:**

No. The limitations/impact discussion is brief and could be expanded.

Suggested additions: simulator–human mismatch risks, judge bias and reward hacking concerns, and practical safety considerations for medical-style recommendations even when evaluated in simulation.

**Strengths And Weaknesses:**

1. Soundness

Strengths:
- The turn-level framing is a practical way to reduce credit-assignment noise while staying aligned with how multi-turn interactions are structured.
- The method is presented to work with multiple advantage estimators and PPO-style objectives, which makes it easier to adopt in existing pipelines.

Weaknesses:
- The empirical evidence relies heavily on LLM user simulators and LLM-based judges, which can affect how well results translate to real user interactions or alternative evaluators.
- Experiments appear focused on relatively short horizons; it is less clear how the approach behaves in longer or more varied conversations.

2. Presentation

Strengths:
- The overall structure (method → training recipe → experiments → analysis) is easy to follow.
- The paper includes analysis beyond main metrics (e.g., correlation/stability checks and a small human label study), which helps interpretation.

Weaknesses:
- Some parts would benefit from clearer separation between what is new in this paper versus what is standard background (implicit reward modeling, PPO details).
- Hyperparameter choices for the normalization variant could be summarized more directly, with a short sensitivity discussion in the main text.

3. Significance

Strengths:
- Multi-turn credit assignment is an important practical issue in interactive RL for LLMs; a simple, reusable turn-wise approach could be useful to others.
- Results are reported across multiple tasks and training setups, which supports broader relevance.

Weaknesses:
- The real-world impact is harder to judge without stronger evidence beyond simulated users and automatic judges (or at least robustness across evaluators).

4. Originality

Strengths:
- The combination of implicit reward shaping with turn-level aggregation is a neat and pragmatic design choice, and the normalization variant is a reasonable extension for stability.

Weaknesses:
- The novelty appears more in packaging and empirical validation than in introducing a fundamentally new learning paradigm; the paper may be best viewed as a solid incremental step.

---

> ### Author Rebuttal · Authors · 2026-03-31
>
> We thank **reviewer URbJ** for their detailed comments and thoughtful suggestions. Below we try to address their questions:
>
> Q1 & W3.1: How robust are the results to different user simulators and different outcome judges?
> ---
> We thank the reviewer for this important question. We try to address this from three aspects:
>
> (1) We further test robustness under a different simulator/judge by replacing Qwen2.5-14B-Instruct with Qwen3-30B-A3B-Instruct-2507. Due to limited resources, we report results on doc writing, with training curves in  [(Sec. 2 in this link)](https://anonymous.4open.science/r/supplementary_figures-3F3B) and results below.
> | Method | BLEU | # Tokens(k) | Score |
> |------------------|------:|------------:|------:|
> | Trajectory-Level | 36.29 | 0.67 | 33.06 |
> | ITPO | 40.93 | 1.08 | 35.51 |
> | Norm-ITPO | 41.40 | 1.07 | 36.00 |
>
> ITPO and Norm-ITPO algorithm still outperform the trajectory-level baseline, showing the robustness against different simulator patterns. We also plan to further include an additional simulator/judge model across the tasks in the final version.
>
> (2) Across all the training experiments, we set the simulator temperature to 1.0 during rollouts to encourage diverse, high-variance user behaviors.
>
> (3) We agree that reviewer just pointed out a promising problem, closing the gap between simulated and real users is an important and interesting open direction.
>
> Q2 & w1.1: Do you have any evidence for transfer to real users, or to evaluation based on human data / human ratings?
> ---
>
> Yes. As described in the *Experiments - Semantic Interpretability*, we construct a validation set with 32 fixed multi-turn rollout trajectories. For each trajectory, three human experts identify the best response and the worst, together with rationale, resulting in 64 labels in total.
>
> We then evaluate the trained implicit RM by comparing its turn-level rewards across the responses within each trajectory, and checking whether its preferences agree with annotations. ITPO and Norm-ITPO achieve accuracies of 47/64 and 48/64, respectively, where a random guess is expected to achieve 16/64 correct selections. Results suggest that implicit PRM, when aggregated at turn level, captures semantic information and produces preferences that are aligned with human judgment.
>
>
> W1.2: Short horizons.
> ---
> We agree with the reviewer that the dialogue horizon is an important design choice. In our experiments, we cap each task interaction at **5–6 turns** for two reasons.
>
> (1) This range aligns with realistic multi-turn problem-solving settings: two human generated benchmarks for multi-turn tasks report their average dialogue lengths of **7.7** and **4** turns, respectively [1,2].
>
> (2) Much longer-horizon interaction is certainly valuable to study, but it introduces several additional challenges, including long-term memory, context window constraints, and user-side consistency. We therefore regard longer-horizon dialogue as an important open direction for future work.
>
> [1] MTRAG: A Multi-Turn Conversational Benchmark for Evaluating Retrieval-Augmented Generation Systems. Katsis, et al. Transactions of the Association for Computational Linguistics, 2025.
>
> [2] MultiVerse: A Multi-Turn Conversation Benchmark for Evaluating Large Vision and Language Models. Lee, et al. ICCV 2025.
>
> W2.1 & W2.2 The clear discussion of the contribution and hyperparameter choices
> ---
> We strongly agree with the reviewer’s suggestion that the paper should more clearly distinguish the background from our novel contributions. Due to space constraints, our current discussion of implicit PRMs is placed in *Appendix A.5* under *More Related Works*. In the final version, we will 1) move this discussion into the main text, 2) provide additional background on PPO, 3) and further clarify the specific contribution of implicit PRMs in our method.
>
> The current hyperparameter choices are listed in *Appendix A.2.2.2* due to space limitation. In the final version, we will add a dedicated paragraph in the main text implementation section to explicitly describe the hyperparameter selection for ITPO.
>
> Limitation
> ---
> We thank the reviewer for this helpful suggestion. We will include a limitation section of the final version.
>
> Specifically, our experiments rely on LLM-based user simulators, which although provide a scalable and controlled environment, yet may not fully capture the diversity and variability of real human users. Therefore, further research on developing more capable user simulators would be valuable. Second, our method depends on LLM judges and learned reward signals, the judge might be biased and the reward improvements may not always translate perfectly to better user-perceived quality. Finally, some tasks considered in this work, such as medical recommendation, are intended primarily as challenging simulated benchmarks for knowledge-intensive interaction. They should not be interpreted as evidence of readiness for real-world high-stakes deployment.

---

> > ### Author Rebuttal · Reviewer_URbJ · 2026-04-03
> >
> > Thanks for authors for providing answers.

---

### Official Review · Reviewer_815G · 2026-03-13

**Soundness:** 3
**Presentation:** 3
**Significance:** 2
**Originality:** 2
**Overall Recommendation:** 4
**Confidence:** 4

**Summary:**

This paper proposes a new method for optimising multi-turn dialogue response generation that addresses the problem of sparse rewards and variation of user behaviour.  The method leverages an Implicit Process Reward Model (IPRM) to derive turn- and token-level rewards from trajectory level outcome rewards, and uses an additional normalisation step to improve training stability.  The method is evaluated on three different tasks, comparing it with a range of baselines.  The results show clear improvements across tasks, based on various metrics.

**Compliance With Llm Reviewing Policy:**

Affirmed.

**Final Justification:**

The rebuttal has addressed my main concerns regarding clarity and the human judgement evaluation.  Based on my assessment of the work's originality and impact, I maintain weak accept (4) as my final recommendation.

**Key Questions For Authors:**

1. Could the authors please provide clearer guidance on how to interpret the diagram in Figure 2?
2. Could the authors please provide a clearer overview of the (task-specific and generic) evaluation metrics used in Section 4?
3. What is the evidence to support the claim that ‘ITPO infers turn-wise preferences that are semantically aligned with human judgment’?

**Limitations:**

It would be good to discuss the limitations of the proposed methodology and/or evaluation, which is not currently included.

**Strengths And Weaknesses:**

Soundness: The proposed methodology is presented in sufficient detail with a clear comparison to existing methods, and the claims are supported by a rigorous empirical evaluation.  However, there is a claim in the abstract about alignment with human judgement, but it is unclear whether that is supported with experimental evidence.

Presentation: The motivation and outline of the method is mostly clear, although I did not quite get how to read the lower part of Fig. 2.  Moreover, in the Experiments section, the definition of the metrics was somewhat lacking/confusing.  For example, in 4.1.1. the Score is not introduced until the Medical Recommendation part, but it is reported for all three tasks.

Significance: RL-based optimisation of LLM-based response generation in dialogue is facing the well-known problem of sparsity of rewards and scalability to long-horizon tasks.  The presented approach seems to provide a well thought out solution to this with fairly strong empirical support.

Originality: The proposed method adapts the IPRM framework to multi-turn dialogue by aggregating token-level rewards to turn-level rewards and applying a normalisation to better redistribute the overall reward.

---

> ### Author Rebuttal · Authors · 2026-03-31
>
> We thank **Reviewer 815G** for their thoughtful comments and questions. Below we try to address their questions:
>
> Q1 & W2.1: Could the authors please provide clearer guidance on how to interpret the diagram in Figure 2?
> ---
>
> The key point is that token-level implicit rewards tend to be *unstable* and *hard to interpret*: The implicit reward assigned to the same token along fixed trajectories can vary substantially across reward model training steps, and high-scoring tokens often lack clear semantic meaning. Fig. 2 illustrates this with a fixed math tutoring initial prompt and its first turn response from a multi-turn rollout trajectory. The figure visualizes the implicit process rewards assigned to each token in response #1 over reward model training, from step 20 to step 370. **Darker colors refer to higher implicit scores, while lighter colors indicate lower scores.**
>
> 1. **Human Preference:** Human judges consider the first-turn response to be the best among all turns, because it concisely requests the missing information.
>
> 2. **Token-level implicit scores:** The token-level scores fluctuate significantly throughout training steps, as highlighted by the red-boxed regions. Moreover, the tokens receiving high scores do not exhibit a clear semantic pattern (e.g, 'the', 'you', 'to'…).
>
> 3. **Turn-level implicit score:** After aggregating token-level scores into a turn-level reward, the first-turn response consistently receives the highest score from step 20 onward. The aggregated signal is more stable.
>
> The revised caption of Figure. 2 is provided in [(Sec.1 in this link)](https://anonymous.4open.science/r/supplementary_figures-3F3B).
>
> Q2 & W2.2:  a clearer overview of the (task-specific and generic) evaluation metrics used in Section 4.
> ---
> We apologize for the confusion regarding the metric definition. The misunderstanding stems from the repeated use of the term *Score* to refer to two different quantities.
>
> 1. **The generic metric** (denoted by **Score S**) is defined as a task-specific reward together with a length regularization term, with the overall objective:  $S = R - \eta \times N$ , where R denotes the task-specific reward and N denotes the response length.
>
> 2. Section 4.1.1 defines the **task-specific reward R** for all the three tasks. In particular, for the medical recommendation setting, the “score” described there refers to R in the equation above, not to the generic metric S.
>
> To avoid this ambiguity, we would rename the terminology in Section 4.1.1 medical recommendation from “score” to "medical reward". We would also clarify the distinction between the task-specific reward R and the generic metric S in Section 4.
>
> Q3 & W1: What is the evidence to support the claim that ‘ITPO infers turn-wise preferences that are semantically aligned with human judgment’?
> ---
> To support the claim, we conduct an experiment, which is described in the *Experiments - Semantic Interpretability* paragraph on page 8. we construct a validation set consisting of 32 fixed multi-turn rollout trajectories. For each trajectory, three human experts identify the best response and the worst response, together with the rationale, resulting in 64 human preference labels in total.
>
> We then evaluate the trained implicit reward model by comparing its aggregated turn-wise reward ranking across the responses within each trajectory, and checking whether its preferences agree with human annotations. The results show that both ITPO and Norm-ITPO achieve accuracies of 47/64 and 48/64, respectively, while random guess is expected to achieve 16/64 correct selection. These results suggest that ITPO could capture meaningful semantic information and produces preferences that are aligned with human judgment.

---

> > ### Author Rebuttal · Reviewer_815G · 2026-04-03
> >
> > Thanks to the authors for their clarifications.

---

### Decision · Program_Chairs · 2026-04-30

**Decision:**

Accept (regular)

**Comment:**

Reviewers were overall positive about this paper. They agreed that sparse and noisy feedback in multi-turn user-LLM interaction is an important problem, and they generally found the turn-level formulation practically useful, especially because the method can be integrated into standard policy-optimization frameworks. Reviewers also considered the empirical results across several tasks to be supportive of the paper's main practical claims.

The remaining concerns were limited and mostly focused on presentation and theory framing. In particular, one reviewer remained only partially satisfied with the theoretical motivation of the normalized variant, while other reviewers asked for clearer explanation of grouping details, metrics, simulator limitations, and the extent of human-judgment alignment. The rebuttal resolved most concerns for three of the four reviewers, and no reviewer raised a major unresolved soundness issue. Still, given that one reviewer retained a meaningful reservation about the theoretical grounding and that the paper's strongest contribution is practical rather than conceptual, I recommend weak accept rather than full accept. The paper appears technically solid and useful, with revision requests mainly around sharpening the motivation and clarifying limitations.